# Graph neural networks and non-commuting operators

**Mauricio Velasco**
Departamento de Informática
Universidad Católica del Uruguay
Montevideo, Uruguay
`mauricio.velasco@ucu.edu.uy`

**Kaiying O'Hare**
Departament of Applied Mathematics and Statistics
Johns Hopkins University
Baltimore, Maryland
`kohare3@jh.edu`

**Bernardo Rychtenberg**
Departamento de Informática
Universidad Católica del Uruguay
Montevideo, Uruguay
`bernardo.rychtenberg@ucu.edu.uy`

**Soledad Villar**
Departament of Applied Mathematics and Statistics
Johns Hopkins University
Baltimore, Maryland
`svillar3@jhu.edu`

## Abstract

Graph neural networks (GNNs) provide state-of-the-art results in a wide variety of tasks which typically involve predicting features at the vertices of a graph. They are built from layers of graph convolutions which serve as a powerful inductive bias for describing the flow of information among the vertices. Often, more than one data modality is available. This work considers a setting in which several graphs have the same vertex set and a common vertex-level learning task. This generalizes standard GNN models to GNNs with several graph operators that do not commute. We may call this model graph-tuple neural networks (GtNN).

In this work, we develop the mathematical theory to address the stability and transferability of GtNNs using properties of non-commuting non-expansive operators. We develop a limit theory of graphon-tuple neural networks and use it to prove a universal transferability theorem that guarantees that all graph-tuple neural networks are transferable on convergent graph-tuple sequences. In particular, there is no non-transferable energy under the convergence we consider here. Our theoretical results extend well-known transferability theorems for GNNs to the case of several simultaneous graphs (GtNNs) and provide a strict improvement on what is currently known even in the GNN case.

We illustrate our theoretical results with simple experiments on synthetic and real-world data. To this end, we derive a training procedure that provably enforces the stability of the resulting model.

## 1 Introduction

Graph neural networks (GNNs) [45, 3, 24, 16] are a widely-used and versatile machine learning tool to process different kinds of data from numerous applications, including chemistry [15], molecular geometry [48], combinatorial optimization [21, 36], among many other. Such networks act on functions on the vertices of a graph (also called signals or vertex features) and use the structure of the graph as a powerful *inductive bias* to describe the natural flow of information among vertices. One of the most common graph neural networks are based on graph convolutions [14], which generalize the notion of message passing. The typical architecture has building blocks which are polynomial functions of the adjacency matrix (or more generally of the shift operator) of a graph composed with componentwise non-linearities. Therefore, such networks implement the idea that the values of a

38th Conference on Neural Information Processing Systems (NeurIPS 2024).

function at a vertex are related with the values at the immediate neighbors of the vertex and also with the values at the neighbors of its neighbors, etc.

Due to the significant practical success and diversity of applications of GNNs, there is a growing interest in understanding their mathematical properties. Researchers have delved into various theoretical aspects of MPNNs, including, for instance, expressivity [35, 47, 9, 1, 8], oversmoothing [44], multi-scale properties [18, 5], and model relaxations [12, 17]. One of the fundamental properties of graph neural networks is their remarkable *transferability property*, which intuitively refers to their ability to perform well in large networks when trained in smaller networks, thus *transfering* knowledge from one to the other. This is in part possible because the number of parameters that defines a GNN is independent of the size of the input graphs. The idea is conceptually related to the algebraic notion of representation stability that has been recently studied in the context of machine learning models [28]. More precisely, if two graphs describe similar phenomena, then a given GNN should have similar repercussions (i.e. similar effect on similar signals) on both graphs. In order to describe this property precisely, it is necessary to place signals and shift operators on different graphs (of potentially different sizes) in an equal footing to allow for meaningful comparisons and to characterize families of graphs describing "similar" phenomena. The seminal work [40] has used the theory of graphons to carry out these two steps, providing a solid theoretical foundation to the transferability properties of GNNs. The theory was further developed in [27, 41, 32, 23, 10], and was extended to other models in [6, 26, 43]. The transferability theory is very related to the stability or perturbation theory of GNNs that studies how GNN outputs change under small perturbations of the graph input or graph signal [42, 7, 13, 22], and conceptually related to the theory of generalization for GNNs [46, 11, 29, 33, 31] though the techniques are different.

In many practical situations a fixed collection of entities serves as common vertices to *several distinct graphs* simultaneously that represent several modalities of the same underlying object. This occurs, for instance, in recommendation systems where the items can be considered as vertices of several distinct similarity graphs. It occurs in the analysis of social networks because individuals often participate in several distinct social/information networks simultaneously and in a wide array of multimodal settings.

The goal of this paper is to extend the mathematical theory of GNNs to account for multimodal graph settings. The most closely related existing work is the algebraic neural network theory of Parada-Mayorga, Butler and Ribeiro [38, 37, 4] who pioneer the use of algebras of non-commuting operators. The setting in this paper could be thought of as a special case of this theory. However, there is a crucial difference: whereas the main results in the articles above refer to the Hilbert-Schmidt norm, we define and analyze block-operator-norms on non-commutative algebras acting on function spaces. This choice allows us to prove stronger stability and transferability bounds that when restricted to classical GNNs improve upon or complement the state-of-the-art theory. In particular, we complement work in [42] by delivering bounds that do not exhibit no-transferable energy, and we complement results in [32] by providing stability bounds that do not require convergence. Our bounds are furthermore easily computable in terms of the networks' parameters improving on the results of [37] and in particular allow us to devise novel training algorithms with stability guarantees.

**Our contributions.** The main contribution of this work is a theoretical analysis for graph neural networks in the multimodal framework where each graph object (or graph tuple) can have several adjacency matrices on a fixed set of vertices. We call this model *graph-tuple neural networks (GtNNs)*. It generalizes GNNs and is naturally suited for taking into account information flows along paths traversing several distinct graphs. This architecture replaces the polynomials $h(X)$ underlying graph convolutional neural networks with non-commutative polynomials $h(X_1, \ldots, X_k)$ on the adjacency matrices of the $k$ graphs in our tuple. More generally our approach via *operator networks* gives a general and widely applicable parametrization for such networks. Our approach is motivated by the theory of switched dynamical systems, where recent algorithmic tools have improved our understanding of the iterative behaviour of non-commuting operators [34]. Our main results are tight stability bounds for GtNNs and GNNs.

The second contribution of this article is the definition of graphon-tuple neural networks (WtNNs) which are the natural limits of (GtNNs) as the number of vertices grows to infinity. Graphon-tuple neural networks provide a good setting for understanding the phenomenon of transferability. Our main theoretical result is a *Universal transferability Theorem* for graphon-graph transference which guarantees that *every* graphon-tuple neural network (without any assumptions) is transferable over

sequences of graph-tuples generated from a given graphon-tuple. This means that whatever a GtNN learns on a graph-tuple with sufficiently many vertices, instantaneously transfers with small error to all other graph-tuples of sufficiently large size provided the graph-tuples we are considering describe a "similar" phenomenon in the sense that they have a common graphon-tuple limit. Contrary to some prior results, under the convergence we consider in this paper, there is no no-transferable energy, meaning that the graphon-graph transferability error goes to zero as the size of the graph goes to infinity.

We show with simple numerical experiments that our theoretical bounds seem tight. In Section 7 we provide experiments on synthetic datasets and a real-world movie recommendation dataset where two graphs are extracted from incomplete tabular data. The stability bounds we obtain are within a small factor of the empirical stability errors. And remarkably, the bounds exhibit the same qualitative behavior as the empirical stability error. In order to perform this experiment we introduce a stable training procedure where linear constraints are imposed during GNN training. The stable training procedure could be considered of independent interest (see, for instance, [7]).

## 2 Preliminary definitions

For an integer $n$ we let $[n] := \{1, 2, \ldots, n\}$. By a *graph $G$ on a set $V$* we mean an undirected, finite graph without self-loops with vertex set $V(G) := V$ and edge set denoted $E(G)$. A *shift matrix* for $G$ is any $|V| \times |V|$ symmetric matrix $S$ with entries $0 \leq S_{ij} \leq 1$ satisfying $S_{ij} = 0$ whenever $i \neq j$ and $(i, j) \notin E(G)$.

Our main object of study will be signals (i.e. functions) on the common vertices $V$ of a set of graphs so we introduce notation for describing them. We denote the *algebra of real-valued functions on the vertex set $V$* by $\mathbb{R}[V]$. Any function $f : V \to \mathbb{R}$ is completely determined by its vector of values so, as a vector space, $\mathbb{R}[V] \cong \mathbb{R}^{|V|}$ however, as we will see later, thinking of this space as consisting of functions is key for understanding the neural networks we consider. Any shift matrix $S$ for $G$ defines a *shift operator* $T_G : \mathbb{R}[V] \to \mathbb{R}[V]$ by the formula $T_G(f)(i) = \sum_{j \in V} S_{ij} f(j)$.

The layers of graph neural networks (GNNs) are built from univariate polynomials $h(x)$ evaluated on the shift operator $T_G$ of a graph composed with componentwise non-linearities. If we have a $k$-tuple of graphs $G_1, \ldots, G_k$ with common vertex set $V$ then it is natural to consider multivariate polynomials evaluated at their shift operators $T_{G_i}$. Because shift operators of distinct graphs generally do not commute this forces us to design an architecture which is parametrized by *noncommutative polynomials*. The trainable parameters of such networks will be the coefficients of these polynomials.

**Noncommutative polynomials.** For a positive integer $k$, let $\mathbb{R}\langle X_1, \ldots, X_k \rangle$ be the *algebra of non-commutative polynomials in the variables $X_1, \ldots, X_k$*. This is the vector space having as basis all finite length words on the alphabet $X_1, \ldots, X_k$ endowed with the bilinear product defined by concatenation on the basis elements. For example in $\mathbb{R}\langle X_1, X_2 \rangle$ we have $(X_1 + X_2)^2 = X_1^2 + X_1 X_2 + X_2 X_1 + X_1^2 \neq X_1^2 + 2 X_1 X_2 + X_2^2$.

The basis elements appearing with nonzero coefficient in the unique expression of any element $h(X_1, \ldots, X_k)$ are called the monomial words of $h$. The degree of a monomial word is its length (i.e. number of letters). For example there are eight monomials of degree three in $\mathbb{R}\langle X_1, X_2 \rangle$, namely: $X_1^3, X_1^2 X_2, X_1 X_2 X_1, X_2 X_1^2, X_2^2 X_1, X_2 X_1 X_2, X_1 X_2^2, X_2^3$. More generally there are exactly $k^d$ monomial words of length $d$ and $\frac{k^{d+1}-1}{k-1}$ monomial words of degree at most $d$ in $\mathbb{R}\langle X_1, \ldots, X_k \rangle$.

Noncommutative polynomials have a fundamental structural relationship with linear operators which makes them suitable for transference. If $W$ is any vector space let $End(W)$ denote the space of linear maps from $W$ to itself. If $T_1, \ldots, T_k \in End(W)$ are any set of linear maps on $W$ then the individual evaluations $X_i \to T_i$ extend to a unique *evaluation homomorphism* $\mathbb{R}\langle X_1, \ldots, X_k \rangle \to End(W)$, which sends the product of polynomials to the composition of linear maps. This relationship (known as universal freeness property) determines the algebra $\mathbb{R}\langle X_1, \ldots, X_k \rangle$ uniquely. This proves that noncommutative polynomials are the only naturally transferable parametrization for our networks. For a polynomial $h$ we denote the linear map obtained from evaluation as $h(T_1, \ldots, T_k)$.

**Operator filters and non-commuting operator neural networks.** Using noncommutative polynomials we will define *operator networks*, an abstraction of both graph and graphon neural networks.

Operator networks will provide us with a uniform generalization to graph-tuple and graphon-tuple neural networks and allow us to describe transferability precisely.

The domain and range of our operators will be powers of a fixed vector space $\mathcal{F}$ of signals. More formally, $\mathcal{F}$ consists of real-valued functions on a fixed domain $V$ endowed with a measure $\mu_V$. The measure turns $\mathcal{F}$ into an inner product space (see [25, Chapter 2] for background) via the formula $\langle f, g \rangle := \int_V fg \, d\mu_V$ and in particular gives it a natural norm $\|f\| := (\langle f, f \rangle)^{\frac{1}{2}}$ which we will use throughout the article. In later sections the set $\mathcal{F}$ will be either $\mathbb{R}[V]$ or the space $L := L^2([0, 1])$ of square integrable functions in $[0, 1]$ but operator networks apply much more generally, for instance to the spaces of functions on a manifold $V$ used in geometric deep learning [2]. By an *operator k-tuple* on $\mathcal{F}$ we mean a sequence $\vec{T} := (T_1, \ldots, T_k)$ of linear operators $T_j : \mathcal{F} \to \mathcal{F}$. The tuple is *nonexpansive* if each operator $T_j$ has norm bounded above by one.

If $h \in \mathbb{R}\langle X_1, \ldots, X_k \rangle$ is a noncommutative polynomial then the *operator filter defined by* $h = \sum_\alpha c_\alpha X^\alpha$ *and the operator tuple* $\vec{T}$ is the linear operator $\Psi(h, \vec{T}) : \mathcal{F} \to \mathcal{F}$ given by the formula

$$h(T_1, \ldots, T_k)(f) = \sum_\alpha c_\alpha X^\alpha(T_1, \ldots T_k)(f)$$

where $X^\alpha(T_1, \ldots, T_k)$ is the composition of the $T_i$ from left to right in the order of the word $\alpha$. For instance if $h(X_1, X_2) := -5X_1 X_2 X_1 + 3X_1^2 X_2$ then the graph-tuple filter defined by $h$ applied to a signal $f \in \mathcal{F}$ is $\Psi(h, T_1, \ldots, T_k)(f) = -5T_1(T_2(T_1(f))) + 3T_1^2(T_2(f))$.

More generally, we would like to be able to manipulate several features simultaneously (i.e. to manipulate vector-valued signals) and do so by building block-linear maps of operators with blocks defined by polynomials. More precisely, if $A, B$ are positive integers and $H$ is a $B \times A$ matrix whose entries are non-commutative polynomials $h_{b,a} \in \mathbb{R}\langle X_1, \ldots, X_k \rangle$ we define the *operator filter determined by* $H$ *and the operator tuple* $\vec{T}$ to be the linear map $\Psi(H, \vec{T}) : \mathcal{F}^A \to \mathcal{F}^B$ which sends a vector $x = (x_a)_{a \in [A]}$ to a vector $(z_b)_{b \in [B]}$ using the formula

$$z_b = \sum_{a \in [A]} h_{b,a}(T_1, \ldots, T_k)(x_a)$$

An *operator neural layer with ReLU activation* is an operator filter composed with a pointwise non-linearity. This composition $\sigma \circ \Psi(H, \vec{T})$ yields a (nonlinear) map $\hat{\Psi}(H, \vec{T}) : \mathcal{F}^A \to \mathcal{F}^B$.

Finally an *operator neural network (ONN)* is the result of composing several operator neural layers. More precisely if we are given positive integers $\alpha_0, \ldots, \alpha_N$ and $N$ matrices $H^{(j)}$ of noncommutative polynomials $H_{b,a}^{(j)} := h_{b,a}^{(j)}(X_1, \ldots, X_k)$ for $(b, a) \in [\alpha_{j+1}] \times [\alpha_j]$ and $j = 0, \ldots, N-1$, the *operator neural network (ONN) determined by* $\vec{H} := (H^{(j)})_{j=0}^{N-1}$ *and the operator tuple* $\vec{T}$ is the composition $\mathcal{F}^{\alpha_0} \to \mathcal{F}^{\alpha_1} \to \cdots \to \mathcal{F}^{\alpha_N}$ where the $j$-th map in the sequence is the operator neural layer with ReLu activation $\hat{\Psi}_j(H^{(j)}, \vec{T}) : \mathcal{F}^{\alpha_j} \to \mathcal{F}^{\alpha_{j+1}}$. We write $\Phi(\vec{H}, \vec{T}) : \mathcal{F}^{\alpha_0} \to \mathcal{F}^{\alpha_N}$ to refer to the full composite function. See Appendix A for a discussion on the trainable parameters and the *transfer* to other $k$-tuples. We conclude the Section with a key instance of operator networks:

**An Example: Graph-tuple neural networks (GtNNs).** Henceforth we fix a positive integer $k$, a sequence $G_1, \ldots, G_k$ of graphs with common vertex set $V$ and a given set of shift operators $T_{G_1}, \ldots, T_{G_k}$. We call this information a *graph-tuple* $\vec{G} := (G_1, \ldots, G_k)$ on $V$.

The *graph-tuple filter* defined by a noncommutative polynomial $h(X_1, \ldots, X_k) \in \mathbb{R}\langle X_1, \ldots, X_k \rangle$ and $\vec{G}$ is the operator filter defined by $h$ evaluated at $\vec{T} := (T_{G_1}, \ldots, T_{G_k})$ denoted $\Psi(h, \vec{T}) : \mathbb{R}[V] \to \mathbb{R}[V]$. Exactly as in Section 2 and using the notation introduced there, we define *graph-tuple filters*, *graph-tuple neural layers with ReLu activation* and *graph-tuple neural networks (GtNN) on the graph-tuple* $\vec{G}$ as their operator versions when evaluated at the tuple $\vec{T}$ above.

## 3   Perturbation inequalities

In this Section we introduce our main tools for the analysis of operator networks, namely *perturbation inequalities*. To speak about perturbations we endow the Cartesian products $\mathcal{F}^A$ with max-norms

$$\|z\|_{\boxed{*}} := \max_{a \in [A]} \|z_a\| \text{ if } z = (z_a)_{a \in [A]} \in \mathcal{F}^A.$$

where the norm $\| \bullet \|$ on the right-hand side denotes the standard $L^2$-norm on $\mathcal{F}$ coming from the measure $\mu_V$ as defined in the previous section. Fix feature sizes $\alpha_0, \ldots, \alpha_N$ and matrices $\vec{H} := (H^{(j)})_{j=0,\ldots,N-1}$ of noncommutative polynomials in $k$-variables of dimensions $\alpha_{j+1} \times \alpha_j$ for $j = 0, \ldots, N-1$ and consider the operator-tuple neural networks $\Phi(\vec{H}, \vec{T}) : \mathcal{F}^{\alpha_0} \to \mathcal{F}^{\alpha_n}$ defined by evaluating this architecture on $k$-tuples $\vec{T}$ of operators on the given function space $\mathcal{F}$. A perturbation inequality for this network is an estimate on the sensitivity (absolute condition number) of the output when the operator-tuple and the input signal are perturbed in their respective norms, more precisely perturbation inequalities are upper bounds on the norm

$$\left\| \Phi\left(\vec{H}, \vec{W}\right)(f) - \Phi\left(\vec{H}, \vec{Z}\right)(g) \right\|_{\boxed{*}} \tag{1}$$

in terms of the input signal difference $\|f - g\|_{\boxed{*}}$ and the operator perturbation size as measured by the differences $\|Z_j - W_j\|_{\mathrm{op}}$. The main result of this Section are perturbation inequalities that depend on easily computable constants, which we call *expansion constants* of the polynomials appearing in the matrices $\vec{H}$, allowing us to use them to obtain perturbation estimates for a given network and to devise training algorithms which come with stability guarantees. A key reason for the success of our approach is the introduction of appropriate norms for computations involving block-operators: If $A, B$ are positive integers and $z = (z_a)_{a \in [A]} \in \mathcal{F}^A$ and $R : \mathcal{F}^A \to \mathcal{F}^B$ is a linear operator then we define

$$\|R\|_{\boxed{\mathrm{op}}} := \sup_{z : \|z\|_{\boxed{*}} \leq 1} \left( \|R(z)\|_{\boxed{*}} \right). \tag{2}$$

If $h \in \mathbb{R}\langle X_1, \ldots, X_k \rangle$ is any noncommutative polynomial then it can be written uniquely as $\sum_\alpha c_\alpha x^\alpha$ where $\alpha$ runs over a finite support set of sequences in the numbers $1, \ldots, k$. For any such polynomial we define a set of $k+1$ *expansion constants* via the formulas

$$C(h) := \sum_\alpha |c_\alpha| \quad \text{and} \quad C_j(h) := \sum_\alpha q_j(\alpha)|c_\alpha| \text{ for } j = 1, \ldots, k$$

where $q_j(\alpha)$ equals the number of times the index $j$ appears in $\alpha$. Our main result is the following perturbation inequality, which proves that expansion constants estimate the perturbation stability of nonexpansive operator-tuple networks (i.e. those which satisfy $\|T_j\|_{\mathrm{op}} \leq 1$ for $j = 1, \ldots, k$).

**Theorem 1.** *Suppose $\vec{W}$ and $\vec{Z}$ are two nonexpansive operator $k$-tuples. For positive integers $A, B$ let $H$ be any $B \times A$ matrix with entries in $\mathbb{R}\langle X_1, \ldots, X_k \rangle$. The operator-tuple neural layer with ReLu activation defined by $H$ satisfies the following perturbation inequality: For any $f, g \in \mathcal{F}^A$ and for $m := \min(\|f\|_{\boxed{*}}, \|g\|_{\boxed{*}})$ we have*

$$\left\| \hat{\Psi}(H, \vec{W})(f) - \hat{\Psi}(H, \vec{Z})(g) \right\|_{\boxed{*}} \leq$$

$$\|f - g\|_{\boxed{*}} \max_{b \in [B]} \left( \sum_{a \in [A]} C(h_{b,a}) \right) + m \max_{b \in [B]} \left( \sum_{a \in [A]} \sum_{j=1}^k C_j(h_{b,a}) \|W_j - Z_j\|_{\mathrm{op}} \right). \tag{3}$$

The proof is in Appendix C. We apply the previous argument inductively to obtain a perturbation inequality for general graph-tuple neural networks by adding the effect of each new layer to the bound. More concretely if $\alpha_0, \ldots, \alpha_N$ denote the feature sizes of such a network and $R_W$ and $R_Z$ denote the network obtained by removing the last layer then

**Corollary 2.** *Let $m := \min\left( \|R_{\vec{W}}(f)\|_{\boxed{*}}, \|R_{\vec{Z}}(g)\|_{\boxed{*}} \right)$. The end-to-end graph tuple neural network satisfies the following perturbation inequality:*

$$\|\Phi(\vec{H}, \vec{W})(f) - \Phi(\vec{H}, \vec{Z})(g)\|_{\boxed{*}} \leq$$

$$\|R_{\vec{W}}(f) - R_{\vec{Z}}(g)\|_{\boxed{*}} \max_{b \in [\alpha_N]} \left( \sum_{a \in [\alpha_{N-1}]} C(h_{b,a}^{(N-1)}) \right) + m \max_{b \in [\alpha_N]} \left( \sum_{a \in [\alpha_{N-1}]} \sum_{j=1}^k C_j(h_{b,a}^{(N-1)}) \|W_j - Z_j\|_{\mathrm{op}} \right).$$

$$\tag{4}$$

Corollary 3 below shows that constraining expansion constants allows us to design operator-tuple networks of depth $N$ whose perturbation stability scales *linearly* with the network depth $N$,

**Corollary 3.** *Suppose $\vec{W}$ and $\vec{Z}$ are two nonexpansive operator $k$-tuples. If the inequality*

$$\max_{b \in [\alpha_{j+1}]} \left( \sum_{a \in [\alpha_j]} C(h_{b,a}^{(j)}) \right) \leq 1$$

*holds for $j = 0, \ldots, N-1$ then for $m := \min(\|f\|, \|g\|)_{\boxed{*}}$ we have:*

$$\|\Phi(\vec{H}, \vec{W})(f) - \Phi(\vec{H}, \vec{Z})(g)\|_{\boxed{*}} \leq \|f - g\|_{\boxed{*}} + m \sum_{d=0}^{N-1} \max_{b \in [\alpha_{d+1}]} \sum_{a \in [\alpha_d]} \sum_{j=1}^{k} C_j(h_{b,a}^{(d)}) \|W_j - Z_j\|_{\text{op}}. \quad (5)$$

## 4 Graphons and graphon-tuple neural networks (WtNNs).

In order to speak about transferability precisely, we have to address two basic theoretical challenges. On one hand we need to find a space which allows us to place signals and shift operators living on different graphs in equal footing in order to allow for meaningful comparisons. On the other hand objects that are close in the natural norm in this space should correspond to graphs describing "similar" phenomena. As shown in [41], both of these challenges can be solved simultaneously by the theory of graphons. A graphon is a continuous generalization of a graph having the real numbers in the interval $[0,1]$ as vertex set. The *graphon signals* are the space $L$ of square-integrable functions on $[0,1]$, that is $L := L^2([0,1])$. In this Section we give a brief introduction to graphons and define *graphon-tuple neural networks (WtNN)*, the graphon counterpart of graph-tuple neural networks. Our first result is Theorem 4 which clarifies the relationship between finite graphs and signals on them and their induced graphons and graphon signals respectively allowing us to make meaningful comparisons between signals on graphs with distinct numbers of vertices. The space of graphons has two essentially distinct natural norms which we define later in this Section and review in Appendix B. Converging sequences under such norms provide useful models for families of "similar phenomena" and Theorem 5 describes explicit sampling methods for using graphons as generative models for graph families converging in both norms.

**Comparisons via graphons.** A *graphon* is a function $W : [0,1] \times [0,1] \to [0,1]$ which is measurable and symmetric (i.e. $W(u,v) = W(v,u)$). A *graphon signal* is a function $f \in L := L^2([0,1])$. The *shift operator* of the graphon $W$ is the map $T_W : L \to L$ given by the formula

$$T_W(f)(u) = \int_0^1 W(u,v) f(v) dv$$

where $dv = d\mu(v)$ denotes the Lebesgue measure $\mu$ in the interval $[0,1]$.

A *graphon-tuple* $W_1, \ldots, W_k$ consists of a sequence of $k$ graphons together with their shift operators $T_{W_i} : L \to L$. Exactly as in Section 2 and using the notation introduced there, we define $(A, B)$ *graphon-tuple filters*, $(A, B)$ *graphon-tuple neural layers with ReLu activation* and *graphon-tuple neural networks (WtNN)* as their operator versions when evaluated at the $k$-tuple $\vec{W} := (T_{W_1}, \ldots, T_{W_k})$.

For instance, if we are given positive integers $\alpha_0, \ldots, \alpha_N$ and matrices $H^{(j)}$ with entries given by noncommutative polynomials $H_{b,a}^{(j)} := h_{b,a}^{(j)} \in \mathbb{R}\langle X_1, \ldots, X_k \rangle$ for $(b,a) \in [\alpha_{j+1}] \times [\alpha_j]$ and $j = 0, \ldots, N-1$, the *graphon-tuple neural network (WtNN) defined by* $\vec{H} := (H^{(j)})_{j=0}^{N-1}$ *and* $\vec{W}$ will be denoted by $\Phi(\vec{H}, \vec{W}) : L^{\alpha_0} \to L^{\alpha_N}$.

Next we focus on the relationship between (finite) graphs and graphons. Our main interest are signals (i.e. functions) on the common vertex set $V$ of all the graphs which we think of as a discretization of the graphon vertex set $[0,1]$. More precisely, for every integer $n$ we fix a collection of $n$ intervals $I_j^{(n)} := [\frac{j-1}{n}, \frac{j}{n})$ covering $[0,1)$ and $n$ vertices $v_j^{(n)} := \frac{2j-1}{2n} \in I_j$ which constitute the set $V^{(n)} \subseteq [0,1]$.

To compare functions on different $V^{(n)}$ we will use an *interpolation* operator $i_n$ and a *sampling* operator $p_n$. The *interpolation operator* $i_n : \mathbb{R}[V^{(n)}] \to L$ extends a set of values at the points of $V^{(n)}$

to a piecewise-constant function in $[0, 1]$ via $i_n(g)(u) := \sum_{i=1}^{n} g(v_i^{(n)}) 1_{I_i^{(n)}}(u)$ where $1_Z(x)$ denotes the $\{0, 1\}$ characteristic function of the set $Z$. The *sampling operator* $p_n : L \to \mathbb{R}[V^{(n)}]$ maps a function $f$ to its conditional expectation with respect to the $I_j$, namely the function $g \in \mathbb{R}[V^{(n)}]$ given by the formula $g(v_j) := \int_{I_j} f(v)dv/\mu(I_j) = n \int_{I_j} f(v)dv$. The sampling and interpolation operators satisfy the identities $p_n \circ i_n = id_{\mathbb{R}[V^{(n)}]}$, and $i_n(p_n(f))$ is the piecewise function which on each interval $I_j$ has constant value equal to the average of $f$ on $I_j$. Note that $i_n \circ p_n(f)$ approaches any continuous function $f$ as $n \to \infty$.

Any graph $G$ with $n$ vertices and shift matrix $S_{ij} \in [0, 1]$ induces a (piecewise constant) graphon $W_G$. The *graphon induced by $G$* is given by the formula

$$W_G(x, y) = \sum_{i=1}^{n} \sum_{j=1}^{n} S_{ij} 1_{I_i^{(n)}}(x) 1_{I_j^{(n)}}(y)$$

The following Theorem clarifies the relationship between the shift operator of a graph and that of its induced graphon and how this basic relationship extends to neural networks. Part (2) will allow us to compare graph-tuple neural networks on different vertex sets by comparing their induced graphon-tuple networks (the proof is in Appendix C).

**Theorem 4.** *For every graph-tuple $G_1, \ldots, G_k$ on vertex set $V^{(n)}$ and their induced graphons $W_j := W_{G_j}$ the equality*

$$T_{W_j} = i_n \circ \frac{T_{G_j}}{n} \circ p_n$$

*holds. Moreover, this relationship extends to networks: given feature sizes $\alpha_0, \ldots, \alpha_N$ and matrices $H^{(j)}$ of noncommutative polynomials having no constant term and of compatible dimensions $\alpha_{j+1} \times \alpha_j$ for $j = 0, \ldots, N-1$ the graphon-tuple neural network $\Phi(\vec{H}, \vec{W}) : L^{\alpha_0} \to L^{\alpha_N}$ and the normalized graph-tuple neural network $\Phi(\vec{H}, \vec{G}/n) : \mathbb{R}[V]^{\alpha_0} \to \mathbb{R}[V]^{\alpha_N}$ satisfy the identity*

$$\Phi(H, \vec{T}) = i_n \circ \Phi(H, \vec{T_G}/n) \circ p_n$$

*where $p_n$ and $i_n$ are applied to vectors componentwise.*

**Graphon norms.** The space of graphons is infinite-dimensional and therefore allows for several norms. In infinite-dimensional spaces it is customary to speak about *equivalent norms*, meaning pairs that differ by multiplication by a constant, but also about the coarser relation of *topologically equivalent* norms (two norms are topologically equivalent if a sequence converges in one if and only if it converges in the other). Here we describe two specific norms of interest and describe explicit mechanisms for producing converging sequences in the operator norm.

The most fundamental norm on graphons is $\|W\|_\square := \left| \sup_{U,V \subseteq [0,1]} \iint_{U \times V} W(u, v)dudv \right|$. Known as the cut norm, its importance stems from the fact that two graphons differing by a small cut norm must have similar induced subgraphs in the sense of the counting Lemma of Lovasz and Szegedi (see [30, Lemma 10.23] for details).

As an analytic object however, the cut norm is often unwieldy, so it is typically bounded via more easily computable norms. More precisely, the space of graphons admits two topologically inequivalent norms represented by the operator and Hilbert-Schmidt norms of graphon shift operators respectively (Example 6 shows that they are indeed inequivalent and why this is important in the present context).

Recall that the operator is defined by $\|T_W\|_{\mathrm{op}} := \sup_{\|f\|,\|g\| \le 1} \left| \int_0^1 \int_0^1 W(u, v)f(u)g(v)dudv \right|$ which is the induced norm of $T_W$ as operator from $L^2([0, 1])$ to $L^2([0, 1])$. It is topologically equivalent to the cut norm (see Appendix B). The Hilbert-Schmidt (HS) norm of $T_W$ is the $\ell^2$-norm of the eigenvalues of $T_W$ or equivalently the norm $\|W\|_{L^2}$ thinking of $W$ as a function in the square.

**Graphons as generative models.** Given a graphon $W$ we explicitly construct families of graphs of increasing size which have $W$ as limit. The family associated to a graphon provides a practical realization of the intuitive idea of a collection of graphs which "represent a common phenomenon".

Explicitly constructing such families is of considerable practical importance since they provide us with a controlled setting in which properties like transferability can be tested experimentally over artificially generated data.

Assume $W(x, y)$ is a given graphon. For every integer $n$ we fix a finite set of equispaced vertices as above and a collection of intervals $I_j := [v_j, v_{j+1})$ for $j = 1, \ldots, n-1$ and $I_n := [0, v_1) \cup [v_n, 1]$. We will produce two kinds of undirected graphs with vertex set $V^{(n)} := \{v_1, \ldots, v_n\}$:

1. A deterministic weighted graph, the *weighted template graph* $H_n$ with vertex set $V^{(n)}$ and shift operator $S(v_i, v_j) := \frac{\iint_{I_i \times I_j} W(x,y)dxdy}{\mu(I_i \times I_j)}$ on the edge $(v_i, v_j)$.

2. A random graph, the *graphon-Erdos-Renyi* graph $G_n$ with vertex set $V^{(n)}$ and shift operator $S(v_i, v_j) \in \{0, 1\}$ sampled from a Bernoulli distribution with probability $W(v_i, v_j)$, which is independent for distinct pairs of vertices.

The main result in this Section is that, under mild assumptions on the function $W$, the weighted template graphs and the random graphon-Erdos-Renyi have induced shift operators converging to $T_W$ in suitable norms (see Appendix C for a proof). Note that the second part of the theorem can be seen as a consequence of the analysis in [32] as well.

**Theorem 5.** *For any positive integer $n$ let $\hat{H}^{(n)}$ and $\hat{G}^{(n)}$ be the graphons induced by $H_n$ and $G_n$. The following statements hold*

1. *If $W$ is continuous then $\|T_W - T_{\hat{H}^{(n)}}\|_{\mathrm{HS}} \to 0$*

2. *If $W$ is Lipschitz continuous then $\|T_W - T_{\hat{G}^{(n)}}\|_{\mathrm{op}} \to 0$ almost surely.*

**Example 6.** *Fix $p \in (0, 1)$ and let $W(x, y) = p$ for $x \neq y$ and zero otherwise. The graphon Erdos-Renyi graphs $G^{(n)}$ constructed from $W$ as in (2) above are precisely the usual Erdos-Renyi graphs. Theorem 5 part (2) guarantees that $\|T_W - T_{\hat{G}^{(n)}}\|_{\mathrm{op}} \to 0$ almost surely so this is a convergent graph family in the operator norm. By contrast we will show that the sequence of $T_{\hat{G}^{(n)}}$ does not converge to $T_W$ in the Hilbert-Schmidt norm by proving that $\|T_{\hat{G}^{(n)}}(x, y) - T_W(x, y)\|_{\mathrm{HS}} > \min(p, 1-p) > 0$ almost surely. To this end note that for every $n \in \mathbb{N}$ and every $(x, y) \in [0, 1]^2$ with $x \neq y$ the difference $|W_{\hat{G}^{(n)}}(x, y) - W(x, y)| \geq \min(p, 1-p)$ since the term on the left is either $0$ or $1$. We conclude that $\|T_{\hat{G}^{(n)}}(x, y) - T_W(x, y)\|_{\mathrm{HS}} = \|W_{\hat{G}^{(n)}}(x, y) - W(x, y)\|_{L^2([0,1]^2)} \geq \min(p, 1-p) > 0$ for every $n$ and therefore the sequence fails to converge to zero almost surely.*

The previous example is important for two reasons. First it shows that the operator and Hilbert-Schmidt norm are not topologically equivalent in the space of graphons. Second, the simplicity of the example shows that for applications to transferability, we should focus on the operator norm. More strongly, it proves that trasferability results that depend on the Hilbert-Schmidt norm are not applicable even to the simplest families of examples, namely Erdos-Renyi graphs.

## 5 Universal transferability

Our next result combines perturbation inequalities for graphon-tuple networks and Theorem 4 which compares graph-tuple networks and their induced graphon-tuple counterparts resulting in a *transferability* inequality. As a corollary of this inequality we prove a universal transferability result which shows that *every* architecture is transferable in a converging sequence of graphon-tuples, in the sense that the transferability error goes to zero as the index of the sequence goes to infinity. This result is interesting and novel even for the case of graphon-graph transferability (i.e. when $k = 1$).

**Theorem 7.** *Let $\vec{W}$ be a graphon-tuple and let $\vec{G}$ be a graph-tuple with equispaced vertex set $V \subset [0, 1]$. Let $H$ be any $B \times A$ matrix with entries in $\mathbb{R}\langle X_1, \ldots, X_k \rangle$ and let $\hat{\Psi}(H, \vec{W}) : L^A \to L^B$ (resp $\hat{\Psi}(H, \vec{G}) : \mathbb{R}[V] \to \mathbb{R}[V]$ ) denote the graphon-tuple neural layer (resp. graph-tuple neural layer) with ReLu activation defined by $H$. If $\hat{G}_i$ denotes the graphon induced y $G_i$ then for every*

$f \in L$ the transferability error *for f satisfies*

$$\left\| \hat{\Psi}(H, \vec{T_W})(f) - i_V \circ \hat{\Psi}\left(H, \frac{1}{|V|}\vec{T_G}\right)(p_V(f)) \right\|_{\boxed{*}} \leq$$

$$\|f - i_V \circ p_V(f)\|_{\boxed{*}} \max_{b \in [B]} \left( \sum_{a \in [A]} C(h_{b,a}) \right) + \|f\| \max_{b \in [B]} \left( \sum_{a \in [A]} \sum_{j=1}^{k} C_j(h_{b,a}) \|T_{W_j} - T_{\hat{G}_j}\|_{\mathrm{op}} \right).$$

We are now able to prove the following *Universal transferability* result:

**Theorem 8.** *Suppose* $G^{(\vec{N})} := (G_1^{(N)}, \dots, G_k^{(N)})$ *is a sequence of graph-tuples having vertex set* $V^{(N)} \subseteq [0,1]$. *If the vertex set is equispaced for every* $N$ *and the sequence converges to a graphon-tuple* $\vec{W}$ *in the sense that* $\|T_{\hat{G}_j^{(N)}} - T_{W_j}\|_{\mathrm{op}} \to 0$ *as* $N \to \infty$ *for* $j = 1, \dots, k$ *then every graphon-tuple neural network on* $\vec{W}$ *transfers. More precisely, for every neural network architecture* $\Phi(\vec{H}, \bullet)$ *and every essentially bounded function* $f \in L^2([0,1])$ *the quantity*

$$\left\| \Phi(\vec{H}, \vec{T_W})(f) - i_{V^{(N)}} \Phi\left(\vec{H}, \frac{1}{|V^{(N)}|}\vec{T_G}\right)(p_{V^{(N)}}(f)) \right\|_{\boxed{*}}$$

*converges to zero as* $N \to \infty$. *Furthermore the convergence is uniform among functions* $f$ *with a fixed Lipschitz constant.*

## 6 Training with stability guarantees

Following our perturbation inequalities (i.e., Theorem 1 and Corollary 3) we propose a training algorithm to obtain a GtNN that enforces stability by constraining all the expansion constants $C(h)$ and $C_j(h)$. Consider a GtNN $\Phi(\vec{H}, \vec{T_G})$ and nonexpansive operator $k$-tuples $\vec{T_G}$. Denote the set of $k + 1$ expansion constants for each layer $d = 0, \dots, N - 1$ as

$$C(H^{(d)}) := \max_{b \in [\alpha_{d+1}]} \sum_{a \in [\alpha_d]} C(h_{b,a}^{(d)}) \quad \text{and} \quad C_j(H^{(d)}) := \max_{b \in [\alpha_{d+1}]} \sum_{a \in [\alpha_d]} h_{b,a}^{(d)} \text{ for } j = 1, \dots, k,$$

and write $\vec{C}(\vec{H}) = (C(H^{(d)}))_{d=0}^{N-1}$ and $\vec{C}_j(\vec{H}) = (C_j(H^{(d)}))_{d=0}^{N-1}$ for $j = 1, \dots, k$. Given $k + 1$ vectors of target bounds $\vec{C} := (C^{(d)})_{d=0}^{N-1}$ and $\vec{C}_j := (C_j^{(d)})_{d=0}^{N-1}$ for $j = 1, \dots, k$, and training data $(x_i, y_i) \in \mathcal{F}^{\alpha_0} \times \mathcal{F}^{\alpha_N}$ for $i \in I$, we train the network by a constrained minimization problem

$$\min_{c} \quad \frac{1}{|I|} \sum_{i \in I} \ell(\Phi(\vec{H}(c), \vec{T_G})(x_i), y_i) \quad \text{s.t.} \quad \vec{C}(\vec{H}(c)) \leq \vec{C}, \quad \vec{C}_j(\vec{H}(c)) \leq \vec{C}_j \text{ for } j = 1, \dots, k,$$

where $\ell(\cdot, \cdot)$ is any nonnegative loss function depending on the task, and $c$ denotes all the polynomial coefficients in the network. If we pick $\vec{C}$ to be an all ones vector (or smaller), by Corollary 3, the perturbation stability is guaranteed to scale linearly with the number of layers $N$.

To approximate the solution of the constrained minimization problem we use a penalty method,

$$\min_{c} \quad \frac{1}{|I|} \sum_{i \in I} \ell(\Phi(\vec{H}(c), \vec{T_G})(x_i), y_i) + \lambda[p(\vec{C}(\vec{H}(c)) - \vec{C}) + \sum_{j=1}^{k} p(\vec{C}_j(\vec{H}(c)) - \vec{C}_j)], \quad (6)$$

where $p(\cdot)$ is a componentwise linear penalty function $p(\vec{C}) = (p(C^{(d)}))_{d=0}^{N-1}$ with $p(C^{(d)}) = \max(0, C^{(d)})$. The stable GtNN algorithm picks a fixed large enough penalty coefficient $\lambda$ and trains the network with local optimization methods.

## 7 Experimental data and numerical results

We perform three experiments[1]: (1) we test the tightness of our theoretical bounds on a simple regression problem on a synthetic dataset consisting of two weighted circulant graphs (see Figure

---

[1]Code available: `https://github.com/Kkylie/GtNN_weighted_circulant_graphs` and `https://github.com/mauricio-velasco/operatorNetworks`

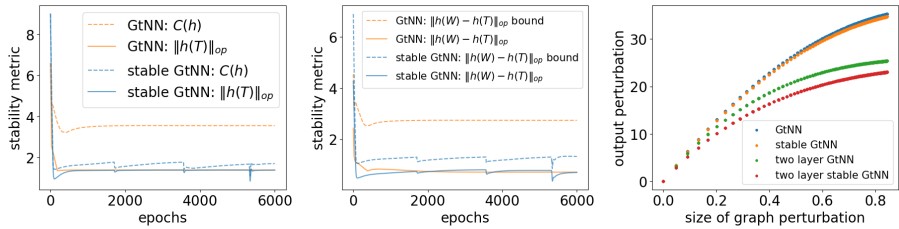

Figure 1: We assess the tightness of our theoretical results on a regression problem on a synthetic data toy example consisting of two weighted circulant graphs. See Appendix D.1 for details. **(Left)** Numerical stability bound $C(h)$ (dashed) and stability metrics $\|h(\vec{T})\|_{\text{op}}$ (solid) with respect to input signal perturbation as a function of the number of epochs for both the standard (1-layer) GtNN (orange) and (1-layer) stable GtNN (blue). **(Middle)** Similar plot for the stability metrics with respect to the graph perturbation $\|h(\vec{W}) - h(\vec{Z})\|_{\text{op}}$ and its upper bound (Lemma 12 part 2 and 3b). For this plot we take $\vec{W} = \vec{T}$, and $\vec{Z}$ is a random perturbation from $\vec{T}$ with $\|Z_1 - W_1\|_{\text{op}} \approx \|Z_2 - W_2\|_{\text{op}} \approx 0.33$. **(Right)** For all four models, compute the 2-norm of the vector of output perturbations from Equation (1) over the test set for various sizes of graph perturbation $(\|T_1 - W_1\|_{\text{op}} + \|T_2 - W_2\|_{\text{op}})/2$, where the additive graph perturbation $T_1 - W_1$ and $T_2 - W_2$ are symmetric matrices with iid Gaussian entries. In addition, each $T_j$ and $W_j$ are normalized such that $\|T_j\|_{\text{op}} \leq 1$ and $\|W_j\|_{\text{op}} \leq 1$ for $j = 1, 2$, so they are nonexpansive operator-tuple networks. **(All)** We observe that adding stability constraints does not affect the prediction performance: the testing R squared value for GtNN is 0.6866, while for stable GtNN is 0.6543.

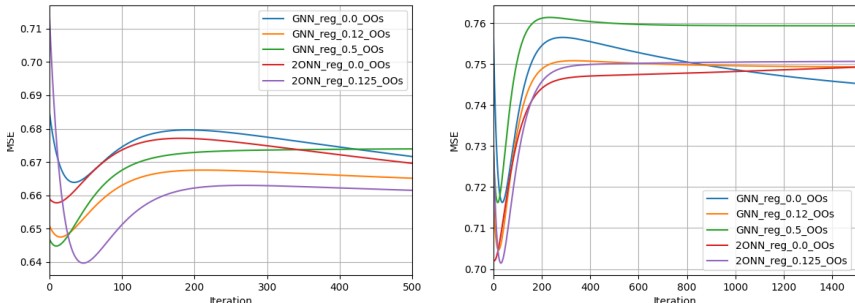

Figure 2: This is an experiment on the MovieLens 100k database, a collection of movie ratings given by a set of 1000 users [39] to 1700 movies. Using collaborative filtering techniques [19] we extract two weighted graphs that we use to predict ratings of movies by user from a held out test set. See details in Appendix D.3. We report the mean squared error (MSE) in the test set as a function of the number of training iterations **(Left)** from 0 to 500 and **(Right)** from 0 to 1500 for the movie recommendation system experiments. We compare the two models GtNN on the tuple of two graphs (2ONN) and GNN on the best single graph between those two (GNN) on various ridge-regularized versions (the legend contains the values of the chosen regularization constants).

1 and Appendix D.1 for details) (2) we assess the transferability of the same model (Appendix D.2), and (3) we run experiments on a real-world dataset of a movie recommendation system where the information is summarized in two graphs via collaborative filtering approaches [19] and it is combined to infer ratings by new users via the GtNN model (see Figure 2 and Appendix D.3).

## 8 Conclusions

In this paper, we introduce graph-tuple networks (GtNNs), a way of extending GNNs to a multi-modal graph setting through the use tuples of non-commutative operators endowed with appropriate block-operator norms. We show that GtNNs have several desirable properties such as stability to perturbations and a universal transfer property on convergent graph-tuples, where the transferability error goes to zero as the graph size goes to infinity. Our transferability theorem improves upon the current state-of-the-art even for the GNN case. Furthermore, our error bounds are expressed in terms of computable quantities from the model. This motivates a novel algorithm to enforce stability during training. Experimental results show that our transferability error bounds are reasonably tight, and that our algorithm increases the stability with respect to graph perturbation. They also suggest that the transferability theorem holds for sparse graph tuples. Finally, the experiments on the movie recommendation system suggest that allowing for architectures based on GtNNs is of potential advantage in real-world applications.

# Acknowledgments

We thank Alejandro Ribeiro for fostering our interest in this topic through various conversations, and Teresa Huang for helpful discussions about theory and code implementation. We thank the organizing committee of the Khipu conference (Montevideo, Uruguay, 2022) for providing a setting leading to the present collaboration. SV is partially supported by NSF CCF 2212457, the NSF–Simons Research Collaboration on the Mathematical and Scientific Foundations of Deep Learning (MoDL) (NSF DMS 2031985), NSF CAREER 2339682, and ONR N00014-22-1-2126. Mauricio Velasco was partially supported by ANII grants FCE-1-2023-1-176172 and FCE-1-2023-1-176242. Bernardo Rychtenberg was partially supported by ANII grant FCE-1-2023-1-176242.

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

# A   Training and transference in operator networks

The *trainable parameters* of an operator network are precisely the coefficients of the noncommutative polynomials involved. For instance, if we have feature sizes $\alpha_0, \ldots, \alpha_N$ and all involved noncommutative polynomials have degree at most $d$ then the number of trainable parameters of the operator network is equal to $\frac{k^{d+1}-1}{k-1} \sum_{i=0}^{N-1} \alpha_i \alpha_{i+1}$. For each choice $\vec{c}$ of such coefficients the network defines a function $\Phi(\vec{H}(c), \vec{T}) : \mathcal{F}^{\alpha_0} \to \mathcal{F}^{\alpha_N}$.

For a fixed collection $\vec{c}$ of coefficients (for instance the one obtained from training on the data $(x_i, y_i)$ for $i \in I$), the resulting polynomials in their respective matrices $H^{(j)}(c)$ allow us to evaluate the trained operator network $\Phi(\vec{H}(c), T_1, \ldots, T_k)$ on any other $k$-tuple of operators $\vec{M} := (M_1, \ldots, M_k)$. Here the $M_j$ are linear maps acting on a vector space of functions $\mathcal{L}$ (possibly different from $\mathcal{F}$) so evaluation defines a new network $\Phi(\vec{H}(c), \vec{M}) : \mathcal{L}^{\alpha_0} \to \mathcal{L}^{\alpha_N}$. Because the coefficients $\vec{c}$ were obtained by training using the operators $\vec{T}$, this network is said to be built by *transference* from $\vec{T}$ to $\vec{M}$.

# B   Graphon norms

The following facts about norms in the space of graphons are well-known:

1. The cut-norm is equivalent to a norm computable from the shift operator. More precisely, define
$$\|W\|_\lozenge := \sup_{\|f\|_\infty, \|g\|_\infty \leq 1} \left| \int_0^1 \int_0^1 W(u,v) f(u) g(v) du dv \right|$$
and note that it this norm is expressible in terms of $T_W$ as $\|W\|_\lozenge = \|T_W\|_{op,\infty,1}$ which is the operator norm of $T_W$ as map from $L^\infty([0,1])$ to $L^1([0,1])$. By [20, Equation 4.4] the cut norm and $\| \bullet \|_\lozenge$ are equivalent because $\|W\|_\square \leq \|W\|_\lozenge \leq 4\|W\|_\square$.

2. The cut-norm and the standard operator norm are topologically equivalent. This can be seen by letting $p \to \infty$ in [20, Lemma E.7, part 1] obtaining the inequality
$$\|W\|_\lozenge \leq \|T_W\|_{op} \leq \sqrt{2}\|W\|_\lozenge^{\frac{1}{2}}$$

3. The elementary inequalities $\|T_W\|_{op} \leq \|T_W\|_{HS}$ and $\|W\|_\lozenge \leq \|W\|_{L^1}$ hold where $\|W\|_{L^1}$ is defined by thinking of $W$ as a function on the square, that is as $\|W\|_{L^1} := \iint_{[0,1]^2} |W(x,y)| dx dy$.

4. The Hilbert-Schmidt norm is topologically equivalent to $\| \bullet \|_{L^1}$. This is implied by the inequality in [20, Lemma E.7, part 2] namely $\|W\|_{L^1} \leq \|T_W\|_{HS} \leq \|W\|_{L^1}^{\frac{1}{2}}$.

# C   Proofs

## C.1   Proof of Theorem 1 (operator-tuple perturbation inequality).

Our first lemma gives a formula for computing the block operator norms introduced in Section 3. Recall that for a positive integer $A$ and $z = (z_a)_{a \in [A]} \in \mathcal{F}^A$ we have $\|z\|_{\boxed{*}} := \max_{a \in [A]} \|z_a\|$

where $\| \bullet \|$ is the norm defined by the measure on $V$ and $\sigma : \mathcal{F}^A \to \mathcal{F}^A$ denotes the componentwise ReLu. Furthermore for positive integers $A, B$ and a linear operator $T : L^A \to L^B$ we have

$$\|T\|_{\boxed{op}} := \sup_{z : \|z\|_{\boxed{*}} \leq 1} \left( \|T(z)\|_{\boxed{*}} \right).$$

**Lemma 9.** *For any linear operator $M : \mathcal{F}^A \to \mathcal{F}^B$ with block-decomposition $(M(z))_b = \sum_{a \in [A]} M_{b,a}(z_a)$ for $b \in [B]$ we have*

$$\|M\|_{\boxed{op}} = \max_{b \in [B]} \left( \sum_{a \in [A]} \|M_{b,a}\|_{op} \right)$$

*Proof.* Assume $\|z\|_{\boxed{*}} \leq 1$. By definition of the norm $\|\bullet\|_{\boxed{*}}$ we have

$$\|M(z)\|_{\boxed{*}} = \max_{b\in[B]} \left\| \sum_{a\in[A]} M_{b,a}(z_a) \right\|$$

By the triangle inequality and the definition of operator norm the last term is bounded by

$$\max_{b\in[B]} \sum_{a\in[A]} \|M_{b,a} z_a\| \leq \max_{b\in[B]} \sum_{a\in[A]} \|M_{b,a}\|_{\mathrm{op}} \|z_a\| \leq \max_{b\in[B]} \left( \sum_{a\in[A]} \|M_{b,a}\|_{\mathrm{op}} \right)$$

where the last inequality follows since $\|z\|_{\boxed{*}} \leq 1$. To prove the equality let $b^*$ be the index for which the sum $\sum_{a\in[A]} \|M_{b,a}\|_{\mathrm{op}}$ achieves the maximum and for $a \in [A]$ let $z_a^*$ be a unit vector in $\mathcal{F}$ with $\|M_{b^*,a}(z_a^*)\| = \|M_{b^*,a}\|_{\mathrm{op}}$. Letting $z^* = (z_a^*)_{a\in[A]}$ we have $\|z^*\|_{\boxed{*}} \leq 1$ and $\|M(z^*)\|_{\boxed{*}} = \max_{b\in[B]} \left( \sum_{a\in[A]} \|M_{b,a}\|_{\mathrm{op}} \right)$ as claimed. $\square$

**Lemma 10.** *The componentwise ReLu is contractive in the $\|\bullet\|_{\boxed{*}}$ norm, that is* $\|\sigma(f) - \sigma(g)\|_{\boxed{*}} \leq \|f - g\|_{\boxed{*}}$ *holds for every $f, g \in \mathcal{F}^A$.*

*Proof.* For any two real valued functions $f_j, g_j$ on any space $V$ the inequality

$$|\max(0, f_j(u)) - \max(0, g_j(u))| \leq |f_j(u) - g_j(u)|$$

holds at every point. Since the left hand side equals the absolute value of a component of $\sigma(f) - \sigma(g)$ the claim is proven by squaring, integrating and taking square roots on both sides and finally maximizing over $j$. $\square$

**Remark 11.** *The previous proof shows that the same conclusion as for ReLu holds for any componentwise non-linearity with the property that its derivative exists almost everywhere and has absolute value uniformly bounded by one.*

The following Lemma summarizes some key inequalities for nonexpansive operator-tuples.

**Lemma 12.** *Assume $\vec{T}, \vec{W}$ and $\vec{Z}$ are non-expansive operator $k$-tuples. The following inequalities hold:*

1. *For every $\alpha \in [k]^d$ we have $\|x^\alpha(\vec{T})\|_{\mathrm{op}} \leq 1$ and for every noncommutative polynomial $h$ we have*
$$\|h(T)\|_{\mathrm{op}} \leq C(h)$$

2. *For every $\alpha \in [k]^d$ we have $\|x^\alpha(\vec{W}) - x^\alpha(\vec{Z})\|_{\mathrm{op}} \leq \sum_{j=1}^k q_j(\alpha)\|W_j - Z_j\|_{\mathrm{op}}$ and for every noncommutative polynomial $h$ we have*
$$\|h(\vec{W}) - h(\vec{Z})\|_{\mathrm{op}} \leq \sum_{j=1}^k C_j(h)\|W_j - Z_j\|_{\mathrm{op}}$$

3. *If $H$ is any $B \times A$ matrix with entries in $\mathbb{R}\langle X_1, \ldots, X_k \rangle$ then:*

   (a) *$\|\Psi(H, \vec{T})\|_{\boxed{\mathrm{op}}} \leq \max_{b\in[B]} \left( \sum_{a\in[A]} C(h_{b,a}) \right)$ and*

   (b) *$\|\Psi(H, \vec{W}) - \Psi(H, \vec{Z})\|_{\boxed{\mathrm{op}}} \leq \max_{b\in[B]} \left( \sum_{a\in[A]} \sum_{j=1}^k C_j(h_{b,a})\|W_j - Z_j\|_{\mathrm{op}} \right).$*

*Proof.* (1) The statement holds for a monomial $x^\alpha$ because operator norms are multiplicative and each $T_j$ has $\|T_j\|_{\mathrm{op}} \leq 1$ by nonexpansivity. The claim for $h(x) = \sum c_\alpha x^\alpha$ follows from the triangle

inequality. (2) First, for any two bounded linear operators $T_A, T_B : \mathcal{F} \to \mathcal{F}$ and any two signals $f, g$ the triangle inequality implies that

$$\|T_A(f) - T_B(g)\| \leq \|T_A\|_{\text{op}}\|f - g\| + \|T_A - T_B\|_{\text{op}}\|g\|.$$

In particular, for any two nonexpansive operators $T_A, T_B$ we have

$$\|T_A(f) - T_B(g)\| \leq \|f - g\| + \|T_A - T_B\|_{\text{op}} \min(\|g\|, \|f\|)$$

Applying this observation inductively to $\vec{Z}, \vec{W}$, any two signals $f, g$ and any word $\alpha \in [k]^d$ we have

$$\left\| x^\alpha(\vec{W})(f) - x^\alpha(\vec{Z})(g) \right\| \leq \|f - g\| + \min(\|f\|, \|g\|) \sum_{j=1}^k q_j(\alpha)\|W_j - Z_j\|_{\text{op}}$$

where $q_j(\alpha)$ is the number of times the index $j$ appears in the word $\alpha$. Setting $f = g$ to be any signal with $\|f\| \leq 1$ we conclude

$$\|x^\alpha(\vec{W}) - x^\alpha(\vec{Z})\|_{\text{op}} \leq \sum_{j=1}^k q_j(\alpha)\|W_j - Z_j\|_{\text{op}}$$

Combining the previous conclusion with the triangle inequality yields

$$\|h(\vec{W}) - h(\vec{Z})\|_{\text{op}} \leq \sum_{j=1}^k C_j(h)\|W_j - Z_j\|_{\text{op}}$$

for any noncommutative polynomial $h$. (3a) By Lemma 9

$$\|\Psi(H, \vec{T})\|_{\boxed{\text{op}}} = \max_{b \in [B]} \sum_{a \in [A]} \|h_{b,a}(\vec{T})\|_{\text{op}}$$

and the claim follows by applying the upper bound we just proved in part (1). (3b) By Lemma 9

$$\|\Psi(H, \vec{W}) - \Psi(H, \vec{Z})\|_{\boxed{\text{op}}} = \max_{b \in [B]} \sum_{a \in [A]} \|h_{b,a}(\vec{W}) - h_{b,a}(\vec{Z})\|_{\text{op}}$$

and the claim follows from applying the upper bound we just proved in part (2). $\square$

We are now ready to prove the main result of this Section,

*Proof of Theorem 1.* By Lemma 10 we have

$$\left\| \hat{\Psi}(H, \vec{W})(f) - \hat{\Psi}(H, \vec{Z})(g) \right\|_{\boxed{*}} \leq \|\Psi(H, \vec{W})(f) - \Psi(H, \vec{Z})(g)\|_{\boxed{*}}$$

By the triangle inequality the quantity above is bounded by the smallest of

$$\|\Psi(H, \vec{W}) - \Psi(H, \vec{Z})\|_{\boxed{\text{op}}}\|f\|_{\boxed{*}} + \|\Psi(H, \vec{Z})\|_{\boxed{\text{op}}}\|f - g\|_{\boxed{*}} \tag{7}$$

and

$$\|\Psi(H, \vec{W}) - \Psi(H, \vec{Z})\|_{\boxed{\text{op}}}\|g\|_{\boxed{*}} + \|\Psi(H, \vec{W})\|_{\boxed{\text{op}}}\|f - g\|_{\boxed{*}}. \tag{8}$$

The Theorem is proven by applying Lemma 12 part (3) to the operator norms and taking the minimum of the resulting upper bounds. $\square$

**Remark 13.** *We expect the bounds of the previous Theorem to be reasonably tight. To establish a precise result in this direction it suffices to prove that the bounds describe the true behavior in special cases. Consider the case $k = 1$, $n = 1$ assuming $T_V, T_W$ and $f \geq g$ are nonnegative scalars with $0 \leq T_W \leq T_V \leq 1$ (a similar reasoning applies to the case of simultaneously diagonal*

*nonexpansive operator tuples of any size). For a univariate polynomial $h(X) = \sum_{j=0}^{d} h_j X^j$ with nonnegative coefficients we have*

$$|h(T_V)(f) - h(T_V)(g)| = \left(\sum_{j=0}^{d} h_j T_V^j\right)(f - g) \leq \sum_{j=0}^{d} |h_j|(f - g)$$

*with equality when $T_V = 1$ and*

$$|h(T_V)(f) - h(T_W)(f)| = \sum_{j=0}^{d} h_j \left(T_V^j - T_W^j\right) f = \sum_{j=0}^{d} h_j \left(j\bar{v}_{(j)}^{j-1}(T_V - T_W)\right) f \leq C_1(h)|T_V - T_W|f$$

*where the second equality follows from the intermediate value theorem (for some $v_{(j)}$ in the interval $[T_W, T_V]$). This equality shows that $C_1(h)$ is the optimal constant bound since the ratio of the left-hand side by $T_V - T_W$ approaches $C_1(h)$ as $T_V$ and $T_W$ simultaneously approach one.*

## C.2 Proofs of Graphon perturbation Theorems

**Lemma 14.** *The following statements hold:*

1. *For every graphon $W$ the inequality $\|T_W\|_{\mathrm{op}} \leq 1$ holds. As a result, for every $\alpha \in [k]^d$ and any $k$-tuple of graphon shift operators the inequality $\|x^\alpha(T_{W_1}, \ldots, T_{W_k})\|_{\mathrm{op}} \leq 1$ holds.*

2. *For any two bounded linear operators $T_A, T_B : L \to L$ and any two signals $f, g \in L$ we have*
   $$\|T_A(f) - T_B(g)\| \leq \|T_A\|_{\mathrm{op}}\|f - g\| + \|T_A - T_B\|_{\mathrm{op}}\|g\|.$$
   *In particular, for any two graphons $A, B$ and any two signals $X, Y$ we have*
   $$\|T_A(f) - T_B(g)\| \leq \|f - g\| + \|T_A - T_B\|_{\mathrm{op}} \min(\|g\|, \|f\|)$$

3. *For any two $k$-tuples of graphon shift operators $\vec{T_W}, \vec{T_Z}$, any two signals $f, g \in L$ and any word $\alpha \in [k]^d$ we have*

   $$\left\|x^\alpha(\vec{T_W})(f) - x^\alpha(\vec{T_Z})(g)\right\| \leq \|f - g\| + \min(\|f\|, \|g\|)\sum_{j=1}^{k} q_j(\alpha)\|T_{W_j} - T_{Z_j}\|_{\mathrm{op}}$$

   *where $q_j(\alpha)$ is the number of times the index $j$ appears in the word $\alpha$.*

*Proof.* (1) Since the operator norm is bounded above by the Hilbert Schmidt norm we have

$$\|T_W\|_{\mathrm{op}} \leq \left(\int_0^1 \int_0^1 W(u,v)^2 dudv\right)^{\frac{1}{2}}$$

and the right hand side is bounded above by one since every graphon satisfies $W(u,v) \in [0,1]$. The inequality on operator norms of monomial words follows from what we have just proven and the submultiplicativity (i.e. $\|AB\|_{\mathrm{op}} \leq \|A\|_{\mathrm{op}}\|B\|_{\mathrm{op}}$) of operator norms. (2) The triangle inequality implies that

$$\|T_A(f) - T_B(g)\| = \|T_A(f) - T_A(g) + T_A(g) - T_B(g)\| \leq \|f - g\|\|T_A\|_{\mathrm{op}} + \|T_A - T_B\|_{\mathrm{op}}\|g\|$$

The second inequality follows from combining the inequality that we just proved with part $(1)$ and exchanging the roles of $A$ and $B$. (3) We prove the statement by induction on $d \geq 0$. If $d = 0$ then $\alpha = \emptyset$, $x^\alpha$ is the identity and the claimed inequality holds with equality. If $d > 0$ let $j := \alpha(1)$ and let $\beta \in [k]^{d-1}$ be the word obtained from $\alpha$ y removing the first (leftmost) term. By construction the equality $x^\alpha = X_j x^\beta$ holds and therefore

$$\|x^\alpha(T_{A_1}, \ldots, T_{A_k})(f) - x^\alpha(T_{B_1}, \ldots, T_{B_k})(g)\|$$

$$= \left\|T_{A_j} x^\beta(T_{A_1}, \ldots, T_{A_k})(f) - T_{B_j} x^\beta(T_{B_1}, \ldots, T_{B_k})(g)\right\|$$

By the second inequality in part $(2)$ and part $(1)$ this is quantity is bounded above by

$$\|x^\beta(T_{A_1},\ldots,T_{A_k})(f) - x^\beta(T_{B_1},\ldots,T_{B_k})(g)\| +$$

$$\|T_{A_j} - T_{B_j}\|_{\mathrm{op}} \min\left(\|x^\beta(T_{A_1},\ldots,T_{A_k})(f)\|, \|x^\beta(T_{B_1},\ldots,T_{B_k})(g)\|\right)$$

applying part $(1)$ we know this is quantity is bounded above by

$$\|x^\beta(T_{A_1},\ldots,T_{A_k})(f) - x^\beta(T_{B_1},\ldots,T_{B_k})(g)\| + \|T_{A_j} - T_{B_j}\|_{\mathrm{op}} \min\left(\|f\|, \|g\|\right)$$

Applying the induction hypothesis to the first term, because $\beta \in [k]^{d-1}$, we see that this expression is bounded above by

$$\left(\|f - g\| + \min(\|f\|, \|g\|) \sum_{i=1}^{k} q_i(\beta) \|T_{A_i} - T_{B_i}\|_{\mathrm{op}}\right) + \|T_{A_j} - T_{B_j}\|_{\mathrm{op}} \min\left(\|f\|, \|g\|\right)$$

where $q_i(\beta)$ is the number of times the index $i$ appears in the word $\alpha$. For each index $i \in [k]$ we have

$$q_i(\alpha) = \begin{cases} q_i(\beta) & \text{if } i \neq j \\ q_i(\beta) + 1 & \text{if } i = j \end{cases}$$

so we conclude that the above sum equals

$$\|f - g\| + \min(\|f\|, \|g\|) \sum_{i=1}^{k} q_i(\alpha) \|T_{A_i} - T_{B_i}\|_{\mathrm{op}}$$

proving the claimed inequality. $\qquad\square$

**Lemma 15.** *Let $A, B$ be positive integers.*

1. *The componentwise ReLu is contractive in the $\|\bullet\|_{\boxed{*}}$ norm, that is*

   $$\|\sigma(f) - \sigma(g)\|_{\boxed{*}} \leq \|f - g\|_{\boxed{*}} \text{ holds for every } f, g \in L^A.$$

2. *Let $\vec{W}$ and $\vec{Z}$ be two graphon $k$-tuples. If $H$ is any $B \times A$ matrix with entries in $\mathbb{R}\langle X_1, \ldots, X_k \rangle$ then the perturbation of the filter $\|\Psi(H, \vec{T_W}) - \Psi(H, \vec{T_Z})\|_{\boxed{\mathrm{op}}}$ is bounded above by*

   $$\max_{b \in [B]} \left(\sum_{a \in [A]} \sum_{j=1}^{k} C_j(h_{b,a}) \|T_{W_j} - T_{Z_j}\|_{\mathrm{op}}\right)$$

   *and furthermore*

   $$\max\left(\|\Psi(H, \vec{T_W})\|_{\boxed{\mathrm{op}}}, \|\Psi(H, \vec{T_Z})\|_{\boxed{\mathrm{op}}}\right)$$

   *is bounded above by $\max_{b \in [B]} \left(\sum_{a \in [A]} C(h_{b,a})\right)$.*

*Proof.* The Claim follows by applying this inequality to $M := \Psi(H, \vec{T_W})$ and to the difference $M := \Psi(H, \vec{T_W}) - \Psi(H, \vec{T_Z})$ together with the operator norm estimates of Lemma 16. $\qquad\square$

**Lemma 16.** *If $\vec{W} := (W_1, \ldots, W_k)$ and $\vec{Z} := (Z_1, \ldots, Z_k)$ are two operator-tuples then for any two signals $f, g \in L$ the quantity $\left\|h(\vec{T_W})(f) - h(\vec{T_Z})(g)\right\|$ is bounded above by*

$$C(h)\|f - g\| + \min(\|f\|, \|g\|) \sum_{j=1}^{k} C_j(h) \|T_{W_j} - T_{Z_j}\|_{\mathrm{op}}$$

*Proof.* By the triangle inequality, for any $f, g$ the quantity $\left\| h(\vec{T_W})(f) - h(\vec{T_Z})(g) \right\|$ is bounded above by

$$\sum_{\alpha \in [k]^{\leq d}} |c_\alpha| \left\| x^\alpha(\vec{T_W})(f) - x^\alpha(\vec{T_Z})(g) \right\|$$

so the claim follows by applying Lemma 14 part (3), reordering the second sum and using the definitions of the expansion constants $C(h)$ and $C_j(h)$. $\qquad\square$

*Proof of Theorem 7.* Denote by $\vec{T_G}$ be the operators in the graph-tuple $\hat{G}$ and let $\vec{T}_{\hat{G}}$ denote their induced graphon operators. If $f \in \mathcal{F}^A$ is any signal then Theorem 1 implies that the following inequality holds

$$\left\| \hat{\Psi}(H, \vec{T_W})(f) - \hat{\Psi}\left(H, \vec{T}_{\hat{G}}\right)(i_V \circ p_V(f)) \right\|_{\boxed{*}} \leq$$

$$\|f - i_V \circ p_V(f)\|_{\boxed{*}} \max_{b \in [B]} \left( \sum_{a \in [A]} C(h_{b,a}) \right) + \|f\| \max_{b \in [B]} \left( \sum_{a \in [A]} \sum_{j=1}^{k} C_j(h_{b,a}) \|T_{W_j} - T_{\hat{G}_j}\|_{\mathrm{op}} \right).$$

Since the points in $V$ are equispaced, Theorem 4 implies that

$$\hat{\Psi}\left(H, \vec{T}_{\hat{G}}\right)(i_V \circ p_V(f)) = i_V \circ \hat{\Psi}\left(H, \frac{\vec{T_G}}{|V|}\right)(p_V \circ i_V \circ p_V(f)) = i_V \circ \hat{\Psi}\left(H, \frac{\vec{T_G}}{|V|}\right)(p_V(f))$$

where the last equality follows from the fact that $p_V \circ i_V$ equals the identity map for any choice of finite set $V \subseteq [0, 1]$. The proof is completed by substituting this equality in the left-hand side of the previous inequality. $\qquad\square$

*Proof of Theorem 8.* For a positive integer $N$ apply Theorem 7 to the graphon-tuple $\vec{W}$ and the graph-tuple $G^{(\vec{N})}$ inductively for every layer of the given architecture $\vec{H}$. Since the matrices $\vec{H}$ defining our architecture involve only finitely many polynomials the hypothesis $\|T_{G_j^{(N)}} - T_{W_j}\|_{\mathrm{op}} \to 0$ guarantees that the upper bound we obtain converge to zero provided $\|f - i_{V^{(N)}} \circ p_{V^{(N)}}(f)\|_{\boxed{*}}$ converges to zero as $N \to \infty$ or equivalently if the function $f$, or more precisely its components are well approximated by their local averages at the sampling points $V^{(N)}$. This is obviously true for essentially bounded functions and happens uniformly for Lipschitz functions with a common constant proving the claim. $\qquad\square$

### C.3 Proof of Theorem 4.

We begin with the following preliminary Lemma,

**Lemma 17.** *For a positive integer $n$ let $G$ be a graph with vertex set $V^{(n)}$ and let $W_G$ be its induced graphon. The following statements hold:*

1. *If $f \in L$ then the equality $T_{W_G}(f) = i_n \circ \frac{T_G}{n} \circ p_n(f)$ holds. More generally for any polynomial $h$ with zero constant term we have*

$$h(T_W) = i_n \circ h(T_G/n) \circ p_n$$

2. *If $g \in \mathbb{R}[V(G)]$ is any function on the vertices of $G$ and $h(x)$ is any univariate polynomial with zero constant term then*

$$h(T_W)(i_n(g)) = i_n\left(h(T_G/n)(g)\right).$$

3. *If $\bar{\sigma}, \sigma$ denote the componentwise ReLu functions in $L$ and $\mathbb{R}[V]$ respectively then the equality $\bar{\sigma} \circ i_n = i_n \circ \sigma$ holds.*

*Proof.* (1) Recall that for $f \in L$ we have $T_W(f)(x) = \int_0^1 W_G(x, y)f(y)dy$ which equals

$$= \int_0^1 \sum_{i=1}^n \sum_{j=1}^n S_{ij} 1_{I_i^{(n)}}(x) 1_{I_j^{(n)}}(y)f(y)dy =$$

$$= \sum_{i=1}^n \sum_{j=1}^n S_{ij} 1_{I_i^{(n)}}(x) \int_{I_j^{(n)}} f(y)dy =$$

$$= \sum_{i=1}^n \left( \sum_{j=1}^n S_{ij} \mu(I_j^{(n)}) \left( \int_{I_j^{(n)}} f(y)dy / \mu(I_j^{(n)}) \right) \right) 1_{I_i^{(n)}}(x) =$$

$$= \left( i_n \circ \frac{T_G}{n} \circ p_n(f) \right)(x)$$

where the last equality holds by definition of $p_n$ and $i_n$ and because $\mu(I_j^{(n)}) = 1/n$ for all $j$. For the second claim note that both sides are linear operators it suffices to prove the claim when $h(x)$ is a monomial of degree $k \geq 1$. This follows immediately by induction using the fact that $p_n \circ i_n = id_{\mathbb{R}[V^{(n)}]}$ for every $n$. (2) Apply the identity proven in part (1) to the function $i_n(g)$ and use the equality $p_n \circ i_n = id_{\mathbb{R}[V^{(n)}]}$. (3) If $g \in \mathbb{R}[V]$ then

$$\overline{\sigma}(i_n(g)) = \overline{\sigma} \left( \sum_{j=1}^n g(v_j) 1_{I_j^{(n)}}(x) \right) = \sum_{j=1}^n \max(g(v_j), 0) 1_{I_j^{(n)}}(x) = i_n(\sigma(g)).$$

$\square$

**Remark 18.** *If the points of the set $V$ are not equally spaced in $[0, 1]$ then the identity in part (1) above* does not hold. *This is a common misconception appearing in several articles in the literature.*

*Proof of Theorem 4.* The first claim is Lemma 17 part (1). For the second claim apply Lemma 17 inductively on layers. $\square$

### C.4 Proof of the sampling Theorem

*Proof of Theorem 5.* (1) By compactness of the square $[0, 1]^2$ given $\epsilon > 0$ there exists $\delta > 0$ such that for all $n$ sufficiently large, every rectangle $I_i \times I_j$ is entirely contained in balls of radius $\delta$ with the property that $|W(x_i, x_j) - W(a_i, a_j)| < \epsilon$ whenever $(x_i, x_j)$ and $(a_i, a_j)$ are in $I_i \times I_j$. In particular, at every point of each square the function deviates at most $\epsilon$ from its mean on this square proving that $\|W - \hat{H}^{(n)}\|_{L^1} \leq \epsilon$. From the results summarized in Section 4 we conclude that $\|T_W - T_{\hat{H}^{(n)}}\|_{HS} \to 0$ in the operator norm as claimed. (2) For a positive integer $n$ let $B^{(n)}$ be the discretization of the graphon $W$ from the values at $V^{(n)} \times V^{(n)}$ defined by

$$B^{(n)}(x, y) = \sum_{i=1}^n \sum_{j=1}^n W(v_i^{(n)}, v_j^{(n)}) 1_{I_i}(x) 1_{I_j}(y)$$

Via the triangle inequality we estimate $\|W - \hat{G}^{(n)}\|_\diamond$ from above as the sum of $\|W - B^{(n)}\|_\diamond$ and $\|B^{(n)} - \hat{G}^{(n)}\|_\diamond$. The first term satisfies the inequality $\|W - B^{(n)}\|_\diamond \leq \|W - B^{(n)}\|_{L^1}$ and thus goes to zero by continuity of $W$ by the argument from part (1). For the second term $\|B^{(n)} - \hat{G}^{(n)}\|_\diamond$ note that both graphons are constant in the squares $I_j \times I_k$ and therefore

$$\|B^{(n)} - \hat{G}^{(n)}\|_\diamond = \max_{a, b \in \{0, 1\}^n} \left| \sum_{i=1}^n \sum_{j=1}^n a_i b_j \frac{W(v_i^{(n)}, v_j^{(n)}) - S_{ij}^{(n)}}{n^2} \right|$$

where the $S_{ij}^{(n)}$ are independent Bernoulli random variables with success probability $W(v_i^{(n)}, v_j^{(n)})$. Given $\epsilon > 0$, let $A_n$ be the event that $\max_{a, b \in \{0, 1\}^n} \left| \sum_{i=1}^n \sum_{j=1}^n a_i b_j \frac{W(v_i^{(n)}, v_j^{(n)}) - S_{ij}^{(n)}}{n^2} \right| \geq \epsilon$ where

$1/n^2 = \mu(I_i \times I_j)$ for every $i, j$. We will show that the series $\sum_n \mathbb{P}(A_n) < \infty$ concluding by the Borel-Cantelli Lemma that $\|B^{(n)} - \hat{G}^{(n)}\|_\Diamond \leq \epsilon$ for all but finitely many integers $n$. Since $\epsilon > 0$ was arbitrary this proves that $\|B^{(n)} - \hat{G}^{(n)}\|_\Diamond \to 0$ almost surely as claimed. To verify the summability we will use a simple concentration inequality. The probability $\mathbb{P}(A_n)$ equals

$$\mathbb{P}\left\{\bigcup_{a,b\in\{0,1\}^n}\left|\sum_{i=1}^n\sum_{j=1}^n a_ib_j\frac{W(v_i^{(n)},v_j^{(n)})-S_{ij}^{(n)}}{n^2}\right|\geq\epsilon\right\}$$

which is bounded above by

$$\leq\sum_{a,b\in\{0,1\}^n}\mathbb{P}\left\{\left|\sum_{i=1}^n\sum_{j=1}^n a_ib_j\frac{W(v_i^{(n)},v_j^{(n)})-S_{ij}^{(n)}}{n^2}\right|\geq\epsilon\right\}$$

Since each of the summands is the sum of $\leq n^2$ independent Bernoulli random variables shifted by their mean and divided by $n^2$, Bernstein's inequality implies that the sum is bounded above by

$$2^{n+1}e^{\left(-\frac{n^2\epsilon^2}{2(1+\epsilon/3)}\right)}=e^{-\frac{n^2\epsilon^2}{2(1+\epsilon/3)}+(n+1)\log(2)}$$

this quantity is summable by the ratio test proving the claim. From the results summarized in Section 4 we know that $\|\bullet\|_\Diamond$ is topologically equivalent to $\|\bullet\|_{\mathrm{op}}$ and we conclude that $\|T_W - T_{\hat{G}^{(n)}}\|_{\mathrm{op}} \to 0$ proving the theorem. □

**Remark 19.** *It is necessary to add some assumption on $W$ for the conclusions of the Lemma above to hold. For instance if $W$ is just in $L^2([0,1]\times[0,1])$ then it can be modified in the countable, and thus Lebesgue measure zero, set $\bigcup_n\left(V^{(n)}\times V^{(n)}\right)$ without altering the operator $T_W$ making an approximation scheme as suggested above impossible.*

## D   Experimental details and additional experiments

The code for experiments on stability for graph tuples and experiments on transferability for sparse graph tuples are available here:
`https://github.com/Kkylie/GtNN_weighted_circulant_graphs.git`. And the code for experiments on real-world data from a movie recommendation system is available here:
`https://github.com/mauricio-velasco/operatorNetworks.git`.

### D.1   Details of experiments on stability for graph tuples

For this experiments we consider a weighted circulant graph of size $n$ with shift matrix $S^{(l)}$ as

$$S_{ij}^{(l)}=\begin{cases}p, & \text{if } i=j\\(1-p)/2, & \text{if } |i-j| \mod n=l\\0, & \text{otherwise.}\end{cases}$$

We generate our data pair $(x_i, y_i) \in \mathcal{F} \times \mathcal{F}$ with $\mathcal{F} = \mathbb{R}[V]$ as

$$y_i=[0.76S^{(l_2)}S^{(l_1)}+0.33S^{(l_1)}S^{(l_2)}+0.3(S^{(l_1)})^3]x_i+\epsilon_i,$$

where each value of $x_i$ is uniformly distributed between $[0, 1]$, $\epsilon_i \in \mathcal{F}$ is normal distributed with standard deviation $\sigma = 0.1$, and we pick $n = 293$, $p = 0.05$, $l_1 = 1$, and $l_2 = 30$. Noted that both input and output have only one feature, i.e., $\alpha_0 = \alpha_N = 1$. We train our model with 800 training data $I$ and test it on 200 testing data $I_{\text{test}}$. We use MSE loss, and use ADAM with learning rate 0.01, $\beta_1 = 0.9$ and $\beta_2 = 0.999$ to train our models. Running these experiments took a few hours on a regular laptop (just CPU).

Denote $T_1$ and $T_2$ as the shift operator corresponding to $S^{(l_1)}$ and $S^{(l_2)}$ respectively. Recall from the main text (Section 7) that we consider four different models: (i) one layer unconstrained GtNN (i.e., $\lambda = 0$ in (6)), (ii) one layer stable GtNN (with $\lambda = 10$), (iii) two layers GtNN with number of hidden feature $\alpha_1 = 2$, and (iv) two layers unconstrained GtNN (with $\lambda = 10$ and $\alpha_1 = 2$). For all four models, we set the non-commutative polynomial $h(T_1, T_2)$ to be any polynomial of degree at most

$d = 3$. Thus, we have 15 trainable coefficients for both one layer models and 60 for both two layer models. For the stable GtNN model, we constrain the expansion constants to be at most half of the corresponding expansion constants obtained after training the unconstrained model. Specifically, let $\vec{C}(\vec{H}^{(i)})$ and $\vec{C}_j(\vec{H}^{(i)})$ denotes the resulting expansion constants vectors for model (i). Then, we set the constraints for model (ii) to be $\vec{C} = \vec{C}(\vec{H}^{(i)})/2$ and $\vec{C}_j = \vec{C}_j(\vec{H}^{(i)})/2$ for $j = 1, 2$. Similarly, we set the constraints for model (iv) to be $\vec{C} = \vec{C}(\vec{H}^{(iii)})/2$ and $\vec{C}_j = \vec{C}_j(\vec{H}^{(iii)})/2$ for $j = 1, 2$.

Figure 1 shows the empirical stability metrics and the corresponding upper bounds as a function of the number of epochs for the one layer models (i) and (ii).

As we see in the proof of Theorem 1 (equation (7) and equation (8)), the equation $\|\Psi(H, \vec{T})\|_{\boxed{\text{op}}} = \|h(\vec{T})\|_{\text{op}}$ shows the output perturbation due to the perturbation from input signal, and $\|\Psi(H, \vec{W}) - \Psi(H, \vec{Z})\|_{\boxed{\text{op}}} = \|h(\vec{W}) - h(\vec{Z})\|_{\text{op}}$ shows the output perturbation due to the perturbation from the graph. Meanwhile, $C(h)$ and $C_j(h)$ are the expansion constants that we constrained for the stable GtNN model. We note that the upper bounds exhibit the same qualitative behavior as the empirical stability metrics, especially for the stable GtNN model where all the curves drop due to the parameter reaching the boundary of the constraint sets. This suggests that our stability bound is tight, and controlling the expansion constants increase the model stability. In addition, adding stability constraints has no harm on the prediction performance, since the testing R squared value for GtNN is 0.6867, while for stable GtNN is 0.6864.

To demonstrate the improvement on stability by our algorithms, we test all four models on various perturbed graphs (while fixing input signal). As shown in Figure 1(right) the stable GtNNs increases the stability under graph perturbations, especially in the 2-layer model.

### D.2 Experiments on transferability for sparse graph tuples

We test the transferability behavior on the weighted circulant graph model from Appendix D.1. We are motivated by the practical setting where we aim to train a model on a small graphs and evaluate it on larger graphs. We consider a piecewise constant graphon tuple $(W_1, W_2)$ induced from the $n = 300$ circulant graph tuple, and similarly we generate a piecewise constant functions by the interpolation operator $i_n$ for each data point.

Next, we use this graphon and piecewise constant function as a generative model to generate deterministic weighted graphs $(G_1, G_2)$ of size $m \leq n$ as training graphs (normalized by $m$) and to generate training data by the sampling operator $p_m$. Since $||T_{W_j} - T_{\hat{G}_j}||_{\text{op}} \to 0$ as $m \to n$, according to Theorem 7 the transferability error goes to 0 too. To demonstrate this, we train five different models, trained with graphs tuples of fixed size $m = 100, 150, 200, 250, 300$ (respectively) and compare the performance of the testing data with $n = 300$.

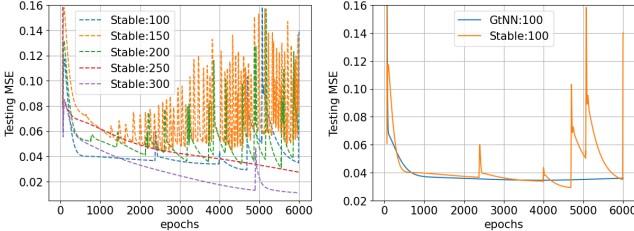

Figure 3: Mean squared error (MSE) on the test set (with testing graph of size $n = 300$) as a function of the number of training epochs for **(Left)** (1-layer) GtNN and **(Middle)** (1-layer) stable GtNN. In both plots we depict the performance of five different models, trained with graphs of sizes $m = 100, 150, 200, 250, 300$ respectively. **(Right)** Comparison of testing MSE between (1-layer) GtNN (blue) and (1-layer) stable GtNN (orange) for training graphs of size $m = 100$ as a function of the number of epochs.

Figure 3 shows that the best testing MSE decreases as the training size $m$ approaches $n$ for the GtNN, which shows transferability holds for sparse graph tuples. For the stable GtNN, the general trend of the testing MSE curves also indicates transferability. In addition, the performance comparison

between GtNN and stable GtNN for $m = 100$ shows that our stability constraint improves the transferability by reducing the best testing MSE. However, this improvement only appears for the $m = 100$ case. All the other cases have worse performance for the stable GtNN. We conjecture this is because the stability constraint makes the training process take a longer time to converge, and whenever it hits the constraint boundaries the MSE jumps, which also makes it harder to converge to a local minimum. It will be interesting to see if other learning algorithms or penalty functions for the stability constraints help improve the performance.

### D.3 Experiments on real-world data from a movie recommendation system

Finally, we present results on the behavior of graph-tuple neural filters on real data as a tool for building a movie recommendation system. We use the publicly available MovieLens 100k database, a collection of movie ratings given by a set of 1000 users [39] to 1700 movies. Our objective is to interpolate ratings among users: starting from the ratings given by a small set of users to a certain movie, we wish to predict the rating given to this movie by all the remaining users. Following [19] we center the data (by removing the mean rating of each user from all its ratings) and try to learn a *deviation from the mean rating function*. More precisely, letting $U$ be the set of users, we wish to learn the map $\phi : \mathbb{R}[U] \to \mathbb{R}[U]$ which, given a partial deviation from the mean ratings function $f : U \to 1, 2, \ldots, 5$ (with missing data marked as zero) produces the full rating function $\hat{f} = \phi(f)$ where $f(u)$ contains the deviation of the mean ratings for user $u$.

The classical Collaborative filtering approach to this problem consists of computing the empirical correlation matrix $B$ among users via their rating vectors. A more recent approach [19] defines a shift operator $S$ on the set of users by sparsifying $B$. More precisely we connect two users whenever their pairwise correlation is among the $k$ highest for both and then approximate $\phi$ as a linear filter or more generally a GNN evaluated on $S$. Although typically superior to collaborative filtering, this approach has a fundamentally ambiguous step: How to select the integer $k$? To the best of our knowledge, there is no principled answer to this question so we propose considering several values simultaneously, defining a tuple of shift operators, and trying to learn $\phi$ via graph-tuple neural networks on $\mathbb{R}[U]$. More specifically we compute two shift operators $T_1, T_2$ by connecting each user to the 10 and 15 most correlated other users respectively, and compare the performance of the GtNN on the tuple $(T_1, T_2)$ (2ONN) with the best between the GNNs on each of the individual operators $T_1$ and $T_2$ (GNN). To make the comparison fair we select the degrees of the allowed polynomials so that all models have the same number of trainable parameters (seven).

Figure 2 (left) shows that an appropriately regularized Graph-tuple network significantly outperforms all other models at any point throughout the first 500 iterations (the minimum occurs when the training error stops improving significantly). However, if the model is over-trained as in the right plot of Figure 2 then it can suffer from a vanishing gradients limitation which may lead to a trained model worse than the best one obtained from standard graph filters. This example suggests that graph-tuple neural networks are of potential relevance to applications.

