# OpenReview forum: "Graph neural networks and non-commuting operators"
_NeurIPS.cc/2024/Conference — NeurIPS 2024 poster_

### Official Review · Reviewer_RRVw · 2024-07-06

**Soundness:** 3
**Presentation:** 4
**Contribution:** 3
**Rating:** 7
**Confidence:** 4

**Summary:**

This paper considers the tasks of extending GNNs to multiple graphs with a shared node set (and the edges representing different types of relations).

In order to do this in a principled manner, the authors consider the algebra of non-commutative polynomials and provide a detailed analysis of stability to perturbations. The authors also extend their setting to graphons and show several consistency results (where the practical implications are to transferability). Notably, the proofs are well written with careful attention to detail and do not suffer from many of the issues common to proofs in ML papers. (See e.g., Remark 16.)

Overall, this paper provides a solid, easily understandable, theoretical framework for an important problem (learning on multigraphs). Acceptance is recommended.

**Strengths:**

Section 6, connecting the stability guarantees to the training procedure is interesting!

Theory is novel, interesting and well explained.

The paper is generally well written and well motivated

The introduction go WtNNs is interesting, well motivated, and provides a solid theoretical framework for transferability of NNs on multigraphs

**Weaknesses:**

It would be helpful if the sections on non-commutative polynomials and operator filters discussed related work on algebraic signal processing and defining neural nets on arbitrary measure spaces, e.g., https://arxiv.org/pdf/2210.17425, https://arxiv.org/pdf/2108.09923, https://arxiv.org/pdf/2210.16272, https://arxiv.org/pdf/2009.01433, https://arxiv.org/abs/2208.08561

Corollaries 2 and 3 are nice, but should probably be combined into a single corollary. Additionally, it looks like equation 5 could be fit onto a single line (possibly with a small amount of negative hspace.)

A more detailed explanation of Example 6 would be helpful. I had to read it several times in order to understand

Having some experiments with real data would be helpful (although I do recognize that the contributions of this paper are primarily theoretical).

A short conclusion would be nice in the camera ready copy.

Line 494: It is not clear where "block operator" norms were introduced

In Lemma 10 (and throughout) the authors should make it more clear that \| \cdot \| refers to the \ell^2 norm since there are many different spaces in this paper.

Lemma 14 is out of place, it should be before the proof of Lemma 14 (which means it would then become Lemma 13)

More details should be added to the proof of theorem 7

Releasing a colab notebook after the review process is okay, but an anonymous github would have been better to improve upon reproducibility

Minor:

Line 91: "behavior than" should be "behavior as"

Line 97: the phrase "loopless" could be confusing. Should make it more clear you mean no self loops

The term End(W) is used without definition. I know what this means, but it would be better to discuss in order to keep the paper self contained for an ML audience.

Line 167: Cartesian should have a capital C. The authors should check throughout for other words derived from last names, e.g., Euclidean or Laplacian

Line 237: V^{(s)} should be V^{(n)}, right?

Line 313: "Corollary" should be "corollary"

Linne 494: "Lemma" should be "lemma"

**Questions:**

Does any of the graphon theory extend to random graphs derived from graphons?

Do you think that analogs of the transferability results for graphon operators can be produced for settings where the graph is derived by subsampling a manifold as in https://arxiv.org/pdf/2305.18467, https://arxiv.org/pdf/2210.00376, https://arxiv.org/pdf/2212.12606, https://arxiv.org/pdf/2307.04056

How does theorem 1 compare with theorem 2 of this paper https://arxiv.org/pdf/2108.09923

Is it possible to define things in any sort of Fourier domain?

**Limitations:**

Yes

---

> ### Author Rebuttal · Authors · 2024-08-05
>
> Thank you for your thorough and positive review. See below our comments and answers to your questions.
>
>
> >Discuss related work ...
>
> We thank the reviewer for pointing this out. We have added all the above references as relevant related work in the Introduction. The new paragraph is in a comment for completeness
>
> >Corollaries 2 and 3 are nice, but should probably be combined into a single corollary.
>
> We believe keeping two corollaries is useful since the first only applies to single blocks and could potentially be used as a step for understanding stability on a different architecture whereas the second one is useful to speak about a large class of examples.
>
> >A more detailed explanation of Example 6 would be helpful. I had to read it several times in order to understand
>
> We have rewritten Example 6 aiming to improve its clarity. We paste it in a comment for completeness.
>
> >Having some experiments with real data would be helpful (although I do recognize that the contributions of this paper are primarily theoretical).
>
> We agree with the reviewers’ comment and have added the results of our initial experiments on Recommendation Systems on real-world  data in a new Final Section entitle “Experiments with real-world data” (see global rebuttal for details). We believe this is a very interesting direction which deserves further exploration and we intend to make it the focus of future work.
>
> >A short conclusion would be nice in the cam ready copy.
> We have added it. Please see Comments for details.
>
> >Line 494: It is not clear where "block operator" norms were introduced
> We have added a reference to the precise Section.
>
> >In Lemma 10 (and throughout) the authors should make it more clear that |  | refers to the \ell^2 norm since there are many different spaces in this paper. We have rewritten the first paragraph of the Section to clarify the meaning of block norms and the fact that on individual components we are referring to the L^2 norm defined by the measure \mu_V
>
> >Lemma 14 is out of place, it should be before the proof of Lemma 14 (which means it would then become Lemma 13)
> We agree and thank the reviewer for pointing this out! We have placed it properly.
>
> >More details should be added to the proof of theorem 7
> We rewrote the proof to make it clearer. The new version is in a comment for completeness.
>
> >Releasing a colab notebook after the review process is okay, but an anonymous github would have been better to improve upon reproducibility
>
> We have added two github repositories containing the scripts used for the article:
>
> https://anonymous.4open.science/r/GtNN_weighted_circulant_graphs-27EB
> https://anonymous.4open.science/r/operatorNetworks-508F
>
> >Does any of the graphon theory extend to random graphs derived from graphons?
>
> Please see rebuttal of point 2 of reviewer (eJ8g) for a general perspective.
>
> > Do you think that analogs of the transferability results for graphon operators can be produced for settings where the graph is derived by subsampling a manifold as [references]?
>
> We believe the answer is yes. Our Theorems imply that whenever one is given a family of converging operators in the rather weak, (and thus widely applicable) operator norm, transferability results hold. In particular it should be possible to use the spectral theory of convergence of laplacian graphs on manifolds (as developed for instance in the interesting work of Calder and Garcia-Trillos [1]) to prove novel transferability results for operator networks operating in this setting.
>
> [1] Jeff Calder, Nicolás García Trillos, Improved spectral convergence rates for graph Laplacians on ε-graphs and k-NN graphs, Applied and Computational Harmonic Analysis, Volume 60,2022.
>
> >How does theorem 1 compare with theorem 2 of this paper https://arxiv.org/pdf/2108.09923
>
> We thank the reviewer for pointing out the above article [denoted PBR]. There are commonalities between the current submission and [PBR] so we have added it as important previous work. Both [PBR] and our article are interested in the stability of filters obtained from representations of operator algebras. The key difference between our stability results and those of [PBR] stem from the choice of norm.
> More specifically: Theorem 2 in [PBR] gives a local perturbation result in the Hilbert-Schmidt norm and contains an unspecified error term. By contrast in our Theorem 1 we give up on the Hilbert-Schmidt norm and instead obtain an upper bound in the operator norm without any unspecified error term. Our result is easily computable, because it depends explicitly on the coefficients of noncommutative polynomials. Our Theorem 1 and [BPR, Theorem 2] are thus complementary inequalities, neither objectively stronger than the other and both could be applied in different circumstances.
> For the kinds of applications we discuss--transferability, the operator norm is a clearly advantageous choice,giving novel Universal transferability results. The fact that our stability bounds are easily computable in terms of polynomial coefficients plays a key role in the novel "stable" training algorithms we propose.
>
> >Is it possible to define things in any sort of Fourier domain?
> We believe the answer is no in general. The Fourier signal processing point of view is basically about diagonalizing the shift operator, however from basic matrix theory we know that a family of operators is simultaneously dagonalizable if and only if it is commutative and each element of the family is diagonalizable so commutativity seems essential. An interesting possible extension is to operator tuples  which are representations of reductive noncommutative groups. However, even for that case the matrices would become only simultaneously block diagonal which would be rather far from the original Fourier transform (although there is a noncommutative Fourier transform that could provide a weak substitute in that case). Nothing like this is available for general noncommuting operators, which are the main focus.

---

> ### Author Response · Authors · 2024-08-06
> **Comments accompanying the rebuttal**
>
> **New paragraph referring to related work in the intro**
>
> The goal of this paper is to extend the mathematical theory of GNNs to account for multimodal graph settings. The most closely related existing work is the algebraic neural network theory of Parada-Mayorga, Butler and Ribeiro [PMR1][PMR2][PMR3] who pioneer the use of algebras of non-commuting operators. The setting in this paper could be thought of as a special case of this theory. However, there is a crucial difference: whereas the main results in the articles above refer to the Hilbert-Schmidt norm, we introduce and analyze block-operator-norms on non-commutative algebras acting on function spaces. This choice allows us to prove stronger stability and transferability bounds that when restricted to classical GNNs improve upon or complement the state-of-the-art theory. In particular, we complement work in \cite{ruiz2021graph} by delivering bounds that do not exhibit no-transferable energy, and we complement results in \cite{maskey2023transferability} by providing stability bounds that do not require convergence. Our bounds are furthermore easily computable in terms of the networks' parameters improving on the results of~\cite{PMR2} and in particular allow us to devise novel training algorithms with stability guarantees.
>
> **New writing of example 6:**
>
> Fix $p\in (0,1)$ and let $W(x,y)=p$ for $x\neq y$ and zero otherwise. The graphon Erdos-Renyi graphs $G^{(n)}$ constructed from $W$ as in $(2)$ above are precisely the usual Erdos-Renyi graphs. Part (2) of the sampling Lemma guarantees that
> $T_W$ converges to $T_{\hat{G}^{(n)}}$ almost surely in the operator norm. By contrast we will show that the sequence  **does not converge** to $T_W$ in the Hilbert-Schmidt norm by proving that $\|T_{\hat{G}^{(n)}}(x,y)-T_W(x,y)\|_{\rm HS}>\min(p,1-p)>0$
>
> almost surely. To this end note that for every $n\in \mathbb{N}$ and every $(x,y)\in [0,1]^2$ with $x\neq y$ the difference $|W_{\hat{G}^{(n)}}(x,y)-W(x,y)|\geq \min(p,1-p)$ since the term on the left is either $0$ or $1$. We conclude that
>
> $\|T_{\hat{G}^{(n)}}(x,y)-T_W(x,y)\|_{\rm HS} = \|W_{\hat{G}^{(n)}}(x,y)-W(x,y)\|_{L^2([0,1]^2)}\geq \min(p,1-p)>0$
>
>  for every $n$ and therefore the sequence fails to converge to zero almost surely.
>
>
> **Short Conclusion (to be added at the end of the article)**
>
> In this paper, we introduce graph-tuple networks, a way of extending GNNs to the multimodal graph setting through the use tuples of non-commutative operators endowed with appropriate block-operator norms.
> We show that GtNNs have several desireable  properties such as stability to perturbations and a Universal transferability property on convergent graph-tuples, where the transferability error goes to zero as the graph size goes to infinity.
> Our transferability theorem improves upon the current state even for the GNN case.
> Furthermore our error bounds are expressed in terms of computable quantities from the model.
> This motivates our novel algorithm to enforce stability during training.
> Experimental results show that our transferability error bounds are reasonably tight, and that our algorithm increases the stability with respect to graph perturbation.
> In addition, they demonstrate that the transferability theorem holds even for sparse graph tuples.
> Finally, the experiments on the movie recommendation system suggest that allowing for architectures based on GtNNs is of potential advantage in real-world applications.

---

> > ### Comment · Reviewer_RRVw · 2024-08-09
> > **Good Job**
> >
> > I thank the authors for their thorough response.

---

### Official Review · Reviewer_hLUB · 2024-07-12

**Soundness:** 3
**Presentation:** 2
**Contribution:** 3
**Rating:** 5
**Confidence:** 3

**Summary:**

This paper introduces a new type of neural network called graph-tuple neural networks (GtNN) that can handle multiple graphs with the same set of nodes and non-commuting graph operators. The authors develop a mathematical theory to show the stability and transferability of GtNNs and derive related bounds, proving a universal transferability theorem for these networks. Their results extend existing transferability theorems for traditional graph neural networks and include experiments on synthetic data to demonstrate their findings.

**Strengths:**

1.	The author put forward a novel theoretical framework GtNN to model a general case when considering transferability and stability, and the Universal transferability Theorem to quantify.
2.	A generative method of mentioned graph sequence is provided instead of merely assuming the existence of the sequence they used in their theoretical framework.
3.	Define the graphon-tuple neural networks (WtNNs) to help theoretical derivation.

**Weaknesses:**

1.	No experiments on real data to prove the transferability and stability gain using their method compared to GNN.
2.	The theoretical background knowledge is not adequately provided.
3.	Lacking proper reasoning phases to connect each sections.

**Questions:**

1.	Can this method be viewed as a training strategy that improves every existing Graph Mining method?
2.	Since WtNNs are “the natural limits of (GtNNs) as the number of vertices grows to infinity”, why did you discuss WtNNs right after the Perturbation inequalities instead of elaborating on GtNNs first.
3.	What is the convergence order, i.e. O(N^k), of the quantity when N goes to infinity?

**Limitations:**

The authors did not adequate discuss the limitation of their work. They may consider talk about how to implement their complicate theories in a understandable way to help this work to gain influence.

---

> ### Author Rebuttal · Authors · 2024-08-05
>
> Thank you for your constructive comments and questions. We address them below:
>
> >No experiments on real data to prove the transferability and stability gain using their method compared to GNN.
>
> We have included some initial results on using operator networks in real-world data in a new final Section of the article (see brief summary in the global rebuttal), we have also added some synthetic transferability experiments. The main contribution of the article is however Theoretical and we have added a proof of tightness of our inequalities in the new Remark 13 in Appendix C as objective proof of the applicability of our results (see rebuttal to Reviewer eJ8g).
>
> >The theoretical background knowledge is not adequately provided.
>
> We have made an effort to add some background knowledge via adding a reference to Hilbert spaces and norms [1] that are needed for a thorough understanding of the article. We also have verified that all novel terms are defined in a rigorous and logically consistent fashion. We kindly ask the reviewer to let us know what background needs further clarification so we can improve the manuscript.
>
> [1] Kostrikin, Alexei I.; Manin, Yuri I. Linear algebra and geometry. Translated from the second Russian (1986) edition by M. E. Alferieff. Revised reprint of the 1989 English edition. Algebra, Logic and Applications, 1997.
>
> >Lacking proper reasoning phases to connect each sections.
>
> We disagree with the reviewer. Each Section begins with clarifying how it connects with the rest of the paper. Due to space limitations it has not been possible to make this more extensive.
>
> >Can this method be viewed as a training strategy that improves every existing Graph Mining method?
>
> In order to apply this method as a training strategy one needs computable bounds which estimate the stability of the resulting network for each choice of the parameters. The main contribution of our work is the discovery of an easily computable form for such bounds: this is the key behind the training strategy. The same method could thus apply to any area where such bounds are available.
>
> >Since WtNNs are “the natural limits of (GtNNs) as the number of vertices grows to infinity”, why did you discuss WtNNs right after the Perturbation inequalities instead of elaborating on GtNNs first.
>
> We agree and thank the reviewer for pointing this out! We have moved the key example of GtNNs to the end of Section 2, which occurs before the definition of graphon, as it should be to facilitate understanding.
>
> >What is the convergence order, i.e. O(N^k), of the quantity when N goes to infinity?
>
> The proof of the Theorem 8 clarifies that the speed of convergence is determined by the speed at which the sequence of induced graphons converges to the limiting  graphon multiplied by a computable constant which depends on the coefficients of the noncommutative polynomials (as in Theorem 7). This quantity is not a fixed N^k, it could vary drastically from one graphon sequence to another.

---

> > ### Comment · Reviewer_hLUB · 2024-08-12
> >
> > Thank you to the authors for your rebuttal. After reading the rebuttal, I decide to maintain my score.

---

### Official Review · Reviewer_eJ8g · 2024-07-14

**Soundness:** 3
**Presentation:** 3
**Contribution:** 3
**Rating:** 6
**Confidence:** 2

**Summary:**

This paper considers the node-level learning task with several graphs sharing the same vertex set called graph-tuple neural networks (GtNNs). The stability and transferability of GtNNs are studied using properties of non-commuting non-expansive operators. The authors show that all GtNNs are transferable on convergent graph-tuple sequences using graphon-tuple neural network limits. They also illustrate the theoretical results with experiments on synthetic data.

**Strengths:**

1. A theoretical framework based on operator networks for analyzing graph neural networks on multimodal data where several graphs share the same set of nodes is proposed. Using this framework, a perturbation inequality, which provides the perturbation stability of non-expansive operator-tuple networks, is proven. The proposed framework may be extended to accommodate other settings and can be used to study other tasks related to GNNs.
2. The authors also define the graphon-tuple neural networks (WtNNs) which are natural limits of GtNNs. Further, a Universal transferability Theorem for graphon-graph transference, which guarantees that the GtNN learned on a graph-tuple with sufficient many vertices transfers to other graph-tuples, is shown. This transferability result is original and addresses the multimodal learning aspect in GNNs.
3. This work contributes to the literature of analyzing GNNs through graph limits and the study of the multimodal setting through operator networks is original. The paper is clearly written.

**Weaknesses:**

1. The tightness of the bounds is only demonstrated through numerical experiments rather than formal theoretical analysis.
2. The results are based on strong assumptions such as the existence of graphons induced by the graphs, the continuity of the graphons, the equispaceness of the vertex set, and the graph-tuple convergence in the operator norm. However, these assumptions may be standard in the literature.
3. The results lack support from real-world experiments.

**Questions:**

1. Can the operator network framework be adapted to the case when the graphs have only partial overlap rather than the same set of vertices?
2. Is it possible to extend the current work to *sparse* graphs when there are no induced graphons?

**Limitations:**

The authors adequately addressed the limitations of the work and clearly stated the assumptions.

---

> ### Author Rebuttal · Authors · 2024-08-05
>
> Thank you for your thoughtful comments and positive feedback. We address each of your comments and questions below:
>
> > The tightness of the bounds is only demonstrated through numerical experiments rather than formal theoretical analysis.
>
> Thank you for pointing this out. We have added a theoretical analysis showing that the bounds are tight for some simultaneously diagonal operator tuples that we reproduce here using the notation of the paper:
>
> **Remark 13 Appendix C**
>
> We expect the bounds of the previous Theorem to be reasonably tight. To establish a precise result in this direction it suffices to prove that the bounds describe the true behavior in special cases. Consider the case $k=1$, $n=1$ assuming $T_V,T_W$ and $f\geq g$ are nonnegative scalars with $0\leq T_W\leq T_V\leq 1$ (a similar reasoning applies to the case of simultaneously diagonal nonexpansive operator tuples of any size). For a univariate polynomial $h(X)=\sum_{j=0}^d h_jX^j$ with nonnegative coefficients we have
> $ |h(T_V)(f)-h(T_V)(g)| =\left(\sum_{j=0}^d h_j T_V^j \right)(f-g)\leq \sum_{j=0}^d |h_j|(f-g)$
> with equality when $T_V=1$ and
>
> $|h(T_V)(f)-h(T_W)(f)| =\sum_{j=0}^d h_j\left(T_V^j-T_W^j\right)f$  =  $\sum_{j=0}^d h_j \left( j v_{(j)}^{j-1} (T_V-T_W) \right) f \leq C_1(h) |T_V-T_W| f$
>
> where the second equality follows from the intermediate value theorem (for some $v_{(j)}$ in the interval $[T_W,T_V]$). This equality shows that $C_1(h)$ is the optimal constant bound since the ratio of the left-hand side by $T_V-T_W$ approaches $C_1(h)$ as $T_V$ and $T_W$ simultaneously approach one.
>
> > The results are based on strong assumptions such as the existence of graphons induced by the graphs, the continuity of the graphons, the equispaceness of the vertex set, and the graph-tuple convergence in the operator norm. However, these assumptions may be standard in the literature.
>
> Yes, they are. The only currently known approach for thinking about transferability is by showing it is a property automatically inherited along convergent graph sequences. Graphons are only one possibility. Crucially, our results show that the comparatively weaker operator norm convergence is sufficient for transferability and this should simplify proving many other transferability results (much of the existing literature focuses on the much stronger –and thus less applicable– Hilbert Schmidt norm convergence).
>
> Furthermore we point out that the perturbation inequalities (stability results) do not require any form of convergence, nor sampling from graphons (equispaced or otherwise). These assumptions are used only for the transferability results. The assumptions we make are not the only possibility, but some assumptions will be needed because in order to establish transferability we need to define what it means to grow the size of graphs consistently (which is equivalent to saying what it means for a sequence of growing size graphs to converge to a limiting distribution). This is needed to have a notion of ground truth to be able to say “if we train a GNN on graphs with $m$ vertices and evaluate it in graphs of $n$ vertices the error is bounded by epsilon” (such a statement relies on an implicit notion of graph convergence).
>
> Here we focus on graphons and the operator norm convergence (and we justify why this convergence is preferable over the Hilbert Schmidt). However, extending our results to other notions of convergence and limiting objects could be of interest. In particular the “Stretched Graphon Model” [1] replaces the notion of graphon $W : [0, 1]^2 → [0, 1]$ by stretching its domain from $[0, 1]$ to $\mathbb R_{+}$  while preserving the integrality condition, to generate sparse graphs. The equispaced sampling in $[0,1]$ is typically replaced by a Poisson sampling in $\mathbb R$. Similarly, the Graphex model introduced in [2] relates to an alternative notion of graph convergence known as sampling convergence [3].
>
> >The results lack support from real-world experiments.
>
> We agree with the reviewers’ comment and have added the results of our initial experiments on Recommendation Systems on real-world  data in a new Final Section “Experiments with real-world data” (see global rebuttal for details). We believe this is a very interesting direction which deserves further exploration and we intend to make it the focus of future work.
>
> >Can the operator network framework be adapted to the case when the graphs have only partial overlap rather than the same set of vertices?
>
> If the union of all vertex sets is called $U$ and we have a shift operator $S$ on a subset $V$ then we can extend the matrix $S$ to a $|U|\times |U|$ matrix $\tilde{S}$ by making all the missing entries zero. In some applications with centered data (for instance in recommendation systems where the scores are centered to have mean zero) this extension by zero is harmless and leads to a satisfactory result. However it should not be used blindly since it could introduce extra unwanted zeroes (blurring the difference between a zero rating and a missing rating)
>
> >Is it possible to extend the current work to sparse graphs when there are no induced graphons?
>
> The perturbation inequality would work over any convergent sequence on the operator norm. As we mention above, even if we don’t use graphons we need a notion of graph limits in order to establish transferability. There are some attempts in the literature to define limiting objects for certain families of sparse graphs [1,2,3] and our theory might immediately apply to those. It would be an interesting research direction to carry this idea forward.
>
> [1] C Borgs, JT Chayes, H Cohn, and N Holden. Sparse exchangeable graphs and their limits via graphon processes. JMLR, 2018.
> [2] V Veitch and DM Roy. The class of random graphs arising from exchangeable random measures, 2015.
> [3] C Borgs, JT Chayes, H Cohn and V Veitch. Sampling perspectives on sparse exchangeable graphs. Ann. Prob., 2019.

---

> ### Comment · Reviewer_eJ8g · 2024-08-12
> **Reply to rebuttal**
>
> I thank the authors for the detailed responses and agreeing to address some of the criticism. I maintain my positive score of the paper.

---

### Author Rebuttal · Authors · 2024-08-05

We thank the reviewers for their thorough reading and constructive comments. We are encouraged that all three reviewers found the paper interesting. The paper’s main weakness, according to the reviews, was the lack of a more thorough empirical evaluation. To address this, we include two additional experiments, one of transferability on a synthetic dataset, and one on a real-world dataset of a movie recommendation system where information from two graphs is combined using the GtNN model. Figures 2 and 3, and references are included in the PDF.

**Experiments on transferability for sparse graph tuples**

We test the transferability behavior on the weighted circulant graph model from Appendix D. We are motivated by the practical setting where we aim to train a model on a small graphs and evaluate it on larger graphs. We consider a piecewise constant graphon tuple $(W_1,W_2)$ induced from the $n=300$ circulant graph tuple, and similarly we generate a piecewise constant functions by the interpolation operator $i_n$ for each data point.

Next, we use this graphon and piecewise constant function as a generative model to generate deterministic weighted graphs $(G_1,G_2)$ of size $m \leq n$ as training graphs (normalized by $m$) and to generate training data by the sampling operator $p_m$.  Since
$||T_{W_j}-T_{\hat{G}_j}|| _{\rm op} \to 0$   as $m\to n$, according to Theorem 7 the transferability error goes to $0$ too. To demonstrate this, we train five different models, trained with graphs tuples of fixed size $m=100,150,200,250,300$ (respectively) and compare the performance of the testing data with $n=300$.

Figure 2 shows that the best testing MSE decreases as the training size $m$ approaches $n$ for the GtNN, which shows transferability holds for sparse graph tuples. For the stable GtNN, the general trend of the testing MSE curves also indicates transferability. In addition, the performance comparison between GtNN and stable GtNN for $m=100$ shows that our stability constraint improves the transferability by reducing the best testing MSE. However, this improvement only appears for the $m=100$ case. All the other cases have worse performance for the stable GtNN. We conjecture this is because the stability constraint makes the training process take a longer time to converge, and whenever it hits the constraint boundaries the MSE jumps, which also makes it harder to converge to a local minimum. It will be interesting to see if other learning algorithms or penalty functions for the stability constraints help improve the performance.

**Experiments on real-world data from a movie recommendation system**

Finally, we present results on the behavior of graph-tuple neural filters on real data as a tool for building a movie recommendation system. We use the publicly available MovieLens 100k database, a collection of movie ratings given by a set of 1000 users [1] to 1700 movies. Our objective is to interpolate ratings among users: starting from the ratings given by a small set of users to a certain movie, we wish to predict the rating given to this movie by all the remaining users. Following [2] we center the data (by removing the mean rating of each user from all its ratings) and try to learn a \emph{deviation from the mean rating function}. More precisely, letting $U$ be the set of users, we wish to learn the map $\phi:\mathbb R[U]\rightarrow\mathbb R[U]$ which, given a partial deviation from the mean ratings function $f:U\rightarrow \{1,2,\dots,5\}$ (with missing data marked as zero) produces the full rating function $\hat{f}=\phi(f)$ where $f(u)$ contains the deviation of the mean ratings for user $u$.

The classical Collaborative filtering approach to this problem consists of computing the empirical correlation matrix $B$ among users via their rating vectors. A more recent approach [2] defines a shift operator $S$ on the set of users by sparsifying $B$. More precisely we connect two users whenever their pairwise correlation is among the $k$ highest for both and then approximate $\phi$ as a linear filter or more generally a GNN evaluated on $S$. Although typically superior to collaborative filtering, this approach has a fundamentally ambiguous step: How to select the integer $k$?  To the best of our knowledge, there is no principled answer to this question so we propose considering several values simultaneously, defining a tuple of shift operators, and trying to learn $\phi$ via graph-tuple neural networks on $\mathbb R[U]$. More specifically we compute two shift operators $T_1,T_2$ by connecting each user to the $10$ and $15$ most correlated other users respectively, and compare the performance of the GtNN on the tuple $(T_1,T_2)$ (2ONN) with the best between the GNNs on each of the individual operators $T_1$ and $T_2$ (GNN). To make the comparison fair we select the degrees of the allowed polynomials so that all models have the same number of trainable parameters (seven).

Figure 3 (left) shows that an appropriately regularized Graph-tuple network significantly outperforms all other models at any point throughout the first $500$ iterations (the minimum occurs when the training error stops improving significantly). However, if the model is over-trained as in the right plot of Figure 3 then it can suffer from a vanishing gradients limitation which may lead to a trained model worse than the best one obtained from standard graph filters. This example suggests that graph-tuple neural networks are of potential relevance to applications.

---

> ### Author Response · Authors · 2024-08-13
> **Question to reviewers**
>
> We thank the reviewers for their time and the constructive and positive feedback we received.
> Since all of them requested experiments on real data, we'd like to ask them whether the experiment on the movies dataset we provide in this global rebuttal satisfies their request.

---

### Author Response · Authors · 2024-08-06
**Anonymous Github repositories**

We have added two anonymous Github repositories containing the scripts used for the article:

https://anonymous.4open.science/r/GtNN_weighted_circulant_graphs-27EB
https://anonymous.4open.science/r/operatorNetworks-508F

---

### Decision · Program_Chairs · 2024-09-25

**Decision:**

Accept (poster)

**Comment:**

This paper introduces graph-tuple neural networks (GtNNs) to handle multiple graphs with shared node sets. The theoretical contributions on stability and transferability of GtNNs are strong and well-developed. All reviewers found the work interesting and technically sound. The main weakness noted was limited experimental evaluation, especially on real-world data. The authors have addressed this in their rebuttal by adding experiments on a movie recommendation dataset. Overall, the novel theoretical framework and analysis represent a valuable contribution. The paper is recommended for acceptance, with the suggestion to incorporate the additional experiments in the final version.